# Learning Chemical Knowledge from Large-Scale Unlabeled Molecular Data for Retrosynthesis

## Abstract

Retrosynthesis, the process of predicting reactants from products, remains a critical challenge in computational chemistry and drug discovery. While recent deep learning methods have shown strong performance, they remain overly reliant on reaction datasets, which are limited in availability and quality. Large-scale unlabeled molecular data encode rich structural patterns that can be leveraged to learn transferable chemical knowledge, but remain largely unexplored. In this work, we propose KnowRetro (Knowledge-Guided Retrosynthesis Prediction), a chemically-aware framework that learns chemical knowledge from large-scale unlabeled molecules to enhance the accuracy and diversity of retrosynthesis prediction. Specifically, KnowRetro first builds a hierarchical knowledge graph from millions of unlabeled molecules, which captures transformation-relevant relationships among molecules, substructures, and functional groups. It then employs chemically guided pre-training based on substructure decomposition to encourage the model to capture fundamental reaction patterns, followed by fine-tuning with a KG adapter designed to inject task-relevant knowledge into reactant generation. Extensive experiments demonstrate that KnowRetro achieves high accuracy with improved robustness and diversity in reactant generation. Our code is available at https://anonymous.4open.science/r/KnowRetro-9C5A.

## 1 Introduction

Retrosynthesis is a foundational methodology in synthesis planning that deconstructs complex molecules into simpler precursors (Corey & Cheng, 1989), enabling pathway design and advancing computational chemistry and drug discovery (Blakemore et al., 2018; Szymkuć et al., 2016).

Traditional retrosynthesis methods rely on expert intuition and predefined rules (Corey et al., 1985; Grzybowski et al., 2018). While effective for well-known or similar compounds, these approaches face challenges in navigating the vast and intricate chemical space (Boström et al., 2018), limiting their adaptability to diverse reactions. The advent of artificial intelligence (AI) and the availability of reaction datasets (Lowe, 2012) have accelerated the shift toward data-driven retrosynthesis. Early works focused on automating the extraction and application of reaction templates (Coley et al., 2017; Segler & Waller, 2017; Dai et al., 2019; Chen & Jung, 2021; Xie et al., 2023), which improved the scalability and interpretability of retrosynthesis. Building upon these advances, data-driven methods further leverage deep learning to directly learn chemical transformations from reaction data. Sequence-to-sequence models (Karpov et al., 2019; Zheng et al., 2019; Tetko et al., 2020; Zhong et al., 2022; Wan et al., 2022; Han et al., 2024) and graph-to-sequence models (Tu & Coley, 2022; Zeng et al., 2024) have shown strong flexibility and predictive performance. However, the chemical knowledge captured by these approaches remains largely implicit and constrained by the scope of available reaction datasets, limiting their ability to capture underlying reaction principles. To mitigate this limitation, recent works augment data-driven models with explicit reaction-level supervision. Reaction-center prediction provides atom- and bond-level supervision at transformation sites (Yan et al., 2020; Shi et al., 2020; Wang et al., 2021; Sacha et al., 2021; Zhong et al., 2023; Wang et al., 2023; Chen et al., 2023; Zhao et al., 2025), while reaction-type classification (Jiang et al., 2023) imposes global constraints that capture characteristic reaction patterns. By incorporating such reaction-specific signals, these methods improve predictive performance in retrosynthesis,

but the supervision they rely on remains restricted to labeled reaction datasets, which inherently limits their applicability.

Based on the above discussion, most existing retrosynthesis methods rely primarily on reaction datasets, which are limited in availability and quality, leaving the structural knowledge in unlabeled molecular data largely unexplored. As illustrated in Figure 1a, structural relations among molecules, substructures, and functional groups can capture plausible transformation pathways even without explicit reaction annotations, highlighting the potential of unlabeled molecular data to support knowledge-driven retrosynthesis. To bridge this gap, we introduce KnowRetro, a

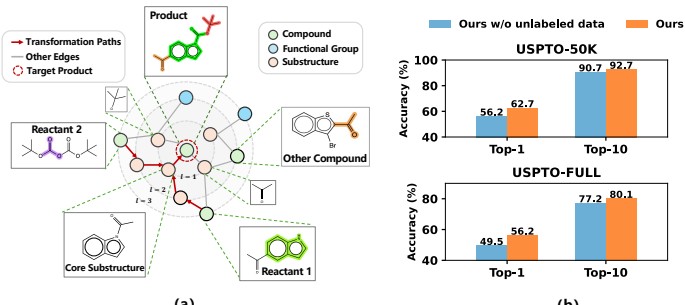

Figure 1: (a) An illustration of the Chemical KG from unlabeled molecules showing compounds, substructures, and functional groups linked by transformation paths. (b) KnowRetro with unlabeled knowledge outperforms its variant without it on USPTO-50K and USPTO-FULL.

chemically-aware framework that systematically learns chemical knowledge from large-scale unlabeled molecules to guide retrosynthesis prediction. Specifically, KnowRetro constructs a hierarchical chemical knowledge graph (KG) that encodes transformation-relevant relationships among molecules, substructures, and functional groups at multiple levels. To better integrate chemical knowledge, we combine chemically guided pre-training, which captures substructure-level reaction patterns, with a fine-tuning stage where a KG adapter distills task-relevant knowledge for robust reactant prediction. Figure 1b show that KnowRetro consistently improves retrosynthesis accuracy on both USPTO-50K and USPTO-FULL by leveraging knowledge from unlabeled molecules.

The contributions of **KnowRetro** are as follows: (1) We introduce a knowledge-guided retrosynthesis framework that systematically leverages hierarchical chemical knowledge from large-scale unlabeled molecules, offering a complementary knowledge source beyond labeled reaction datasets; (2) We design a knowledge-guided learning paradigm that performs chemically guided pre-training to capture transferable reaction patterns, followed by fine-tuning with a KG adapter that distills task-relevant knowledge for robust prediction; (3) Experiments on benchmark datasets demonstrate that KnowRetro consistently outperforms baselines, achieving superior performance, robustness, and diversity, while capturing key reaction-related knowledge in its learned representations.

## 2 RELATED WORK

**Single-step Retrosynthesis Prediction.** Rule-based methods relied on pre-defined reaction templates, where expert knowledge of atom- and bond-level transformations was explicitly encoded into template libraries. While explicit template encoding provides interpretability and enforces chemically valid transformations, dependence on a fixed library of pre-defined templates from reaction data restricts scalability (Coley et al., 2017; Segler & Waller, 2017; Dai et al., 2019). With the advent of reaction datasets, data-driven methods have become dominant. Sequence-to-sequence approaches (Karpov et al., 2019; Zheng et al., 2019; Tetko et al., 2020; Zhong et al., 2022; Han et al., 2024) formulate retrosynthesis as a translation task, offering scalability but often struggling with invalid predictions and limited diversity. Graph-to-sequence frameworks (Tu & Coley, 2022; Zeng et al., 2024) further exploit molecular graph structures to better preserve chemical context and improve prediction validity. Despite these advances, the chemical knowledge captured by such approaches remains largely implicit and is constrained by the scope of available reaction datasets, limiting their ability to capture underlying reaction principles. Recent works further enhance data-driven models with auxiliary reaction-level supervision, including reaction-center prediction that provides atom- and bond-level signals on transformation sites (Yan et al., 2020; Sacha et al., 2021; Wang et al., 2021; Chen et al., 2023; Zhao et al., 2025; Zhong et al., 2023), and reaction-type classification that imposes global constraints on reaction patterns (Jiang et al., 2023). The performance

Figure 2: Overview of the KnowRetro framework. **(a)** Molecules are decomposed into substructures and functional groups to form a multi-level KG $G_{kg}$, which is then encoded by a hierarchical-knowledge encoder; **(b)** Reaction-aware pretraining via SMILES-to-substructure translation; **(c)** For retrosynthesis, the product-side knowledge retrieved in (a) is distilled into a informative latent representation by a task-relevant knowledge adapter. This representation conditions the reaction-aware encoder from (b), enabling it to produce knowledge-augmented embeddings that guide the reactant decoder in generating reactants. Symbols: $\oplus$ indicates addition; $\odot$ denotes element-wise product.

gains of these strategies demonstrate the value of incorporating auxiliary reaction-level supervision into retrosynthesis, yet they fundamentally depend on annotated reaction datasets, leaving the transferable chemical information embedded in unlabeled molecular data largely underexplored.

**Chemical Knowledge Graphs.** KGs have been successfully applied in fields such as drug repurposing (Pan et al., 2022) and drug-drug interaction (Lin et al., 2020). To introduce fundamental chemical domain knowledge, specialized chemical KGs have been developed. For instance, chemical element-level KGs incorporate properties like electronegativity to represent fundamental molecular characteristics (Fang et al., 2022). Similarly, reaction-level KGs represent molecular transformations by modeling reactants and products as nodes, with reaction templates as edges, providing a structured framework for understanding connectivity changes during reactions (Xie et al., 2024). However, existing KGs often overlook multi-level reaction knowledge, critical for modeling the localized structural changes of chemical reactions. Our work addresses this by constructing a hierarchical chemical KG from unlabeled molecules, capturing relations among molecules, substructures, and functional groups to supply a new source of transferable reaction knowledge.

## 3 PROPOSED METHOD

**Problem Definition.** In this work, we formulate retrosynthesis prediction as a sequence-to-sequence translation task, integrating hierarchical chemical knowledge within an end-to-end learning framework. The goal is to generate a valid set of reactants from a given product molecule. Given a product molecule $x \in \mathcal{M}$, represented as a SMILES sequence $x = [x_1, x_2, \ldots, x_m]$ with $m$ tokens, where $\mathcal{M}$ denotes the molecular space. The model aims to generate a reactant sequence $y^{\text{rxn}} = [y_1, y_2, \ldots, y_n]$, constructed by concatenating the SMILES strings of all reactants using the dot symbol (.) and tokenizing the result into $n$ tokens. Generation is performed in an autoregressive manner, where each token is predicted conditioned on the product and previously generated tokens. The model is trained by minimizing the negative log-likelihood of the ground-truth sequence over the training dataset $\mathcal{D}$:

$$\mathcal{L}_{\text{gen}}(\theta) = - \sum_{(x, y^{\text{rxn}}) \in \mathcal{D}} \sum_{t=1}^{n} \log P_\theta(y_t^{\text{rxn}} \mid y_{<t}^{\text{rxn}}, x), \tag{1}$$

where $\theta$ denotes model parameters. We adopt teacher forcing during training, providing the ground-truth prefix $y_{<t}^{\text{rxn}}$ at each decoding step.

**Overview.** Here, we propose KnowRetro, a knowledge-guided retrosynthesis framework that integrates hierarchical chemical knowledge from large-scale unlabeled molecules with molecular sequence modeling for retrosynthesis prediction. The overall architecture of KnowRetro is shown in Figure 2, which is divided into three modules: (a) Given large-scale unlabeled molecules, hierarchical chemical knowledge is constructed by decomposing molecules into substructures and

functional groups, forming a multi-level chemical knowledge graph encoded by a hierarchical-knowledge encoder (Section 3.1); (b) We pretrain a reaction-aware encoder–decoder via a SMILES-to-substructure translation objective designed to recover chemically meaningful fragments from input molecules. This pretraining stage enables the model to capture general fragment-level reactivity patterns; (c) For the downstream retrosynthesis task, product knowledge retrieved in (a) is distilled into a informative latent representation by a task-relevant knowledge adapter, which conditions the reaction-aware encoder from (b). The knowledge-augmented encoder representation is then used by a reactant decoder to generate reactants (Section 3.3).

## 3.1 HIERARCHICAL CHEMICAL KNOWLEDGE GRAPH REPRESENTATION

**Hierarchical Knowledge Graph Construction.** Unlike previous works that primarily focused on constructing knowledge graphs from chemical elements (Fang et al., 2022) (e.g., attributes from the Periodic Table) and reaction-level data (Xie et al., 2024) (e.g., representing reactants and products as nodes with templates as relations), we construct a hierarchical chemical knowledge graph from large-scale unlabeled molecules, capturing hierarchical relationships among molecules, substructures, and functional groups to model structural transformations in chemical reactions. Specifically, we decompose molecules into meaningful substructures and functional groups using chemically guided algorithms. BRICS decomposition (Degen et al., 2008) generates synthetically relevant building blocks by breaking molecular bonds, $x \xrightarrow{\text{BRICS}} \{s_1, s_2, \ldots, s_k\}$, where $x \in \mathcal{M}$ denotes molecule, and $\{s_1, s_2, \ldots, s_k\}$ represents the substructures obtained from it. The number of substructures, $k$, depends on the molecular complexity, with each substructure $s_i$ preserving chemically meaningful and synthetically relevant fragments. Functional groups are identified using SMARTS-based matching with RDKit tool (Landrum et al., 2013), $s_i \xrightarrow{\text{SMARTS}} \{fg_1, fg_2, \ldots, fg_t\}$, where $\{fg_1, fg_2, \ldots, fg_t\}$ are the functional groups identified within substructure $s_i$, and $t$ denotes the number of identified groups, ensuring accurate detection of key reactive sites. The resulting chemical knowledge graph $\mathcal{G}_{\text{kg}}$ encodes hierarchical relationships among molecules, substructures, and functional groups. Formally, it is defined as $\mathcal{G}_{kg} = (\mathcal{V}, \mathcal{E}, \mathcal{R})$, where $\mathcal{V}$ denotes nodes, including molecules, substructures, and functional groups; $\mathcal{E}$ represents the edges capturing their relationships; and $\mathcal{R}$ specifies relation types, such as *has_substructure* and *has_funcgroup*. This KG serves as a foundation for capturing reaction patterns through hierarchical structural relationships.

**Encoding of Hierarchical Knowledge.** To capture the structural and relational knowledge within the KG, we employ a 2-layer Relational Graph Convolutional Network (RGCN) (Schlichtkrull et al., 2018), which incorporates relation types into message passing to capture hierarchical chemical dependencies. The node update process at each layer is defined as:

$$\mathbf{e}_i^{(l+1)} = \sigma \left( \sum_{rel \in \mathcal{R}} \sum_{j \in \mathcal{N}_i^{rel}} \frac{1}{c_{i,rel}} \mathbf{W}_{rel}^{(l)} \mathbf{e}_j^{(l)} + \mathbf{W}_o^{(l)} \mathbf{e}_i^{(l)} \right), \tag{2}$$

where $\mathbf{e}_i^{(l)}$ and $\mathbf{e}_j^{(l)}$ denote the embeddings of node $i$ and its neighbor $j$ at layer $l$; $\mathbf{W}_{rel}^{(l)}$ is the relation-specific transformation matrix for relation $rel \in \mathcal{R}$; $\mathcal{N}_i^{rel}$ denotes the set of neighbors of node $i$ under relation $rel$; $c_{i,rel} = |\mathcal{N}_i^{rel}|$ is a normalization factor; $\mathbf{W}_o^{(l)}$ is the self-loop transformation for node $i$; and $\sigma(\cdot)$ is a nonlinear activation function (e.g., ReLU). The optimization objective is:

$$\mathcal{L}_{\text{kg}} = -\frac{1}{|E|} \sum_{(head, rel, tail, y^{\text{kg}}) \in E} \Big[ y^{\text{kg}} \log \big( \text{sigmoid}(f(head, rel, tail)) \big)$$

$$+ (1 - y^{\text{kg}}) \log \big( 1 - \text{sigmoid}(f(head, rel, tail)) \big) \Big] \tag{3}$$

where $E$ includes both positive triples from $\mathcal{E}$ and negative triples generated via negative sampling. The binary label $y^{kg} \in \{0, 1\}$ indicates whether a triple is valid ($y^{kg} = 1$) or corrupted ($y^{kg} = 0$). The scoring function follows the DistMult formulation (Yang et al., 2014), $f(head, rel, tail) = \mathbf{e}_{head}^\top R_{rel} \mathbf{e}_{tail}$, where $R_{rel} \in \mathbb{R}^{d \times d}$ is a diagonal matrix specific to relation $rel$, and $\mathbf{e}_{head}, \mathbf{e}_{tail} \in \mathbb{R}^d$ are the embeddings of the head and tail entities, respectively. This objective encourages high scores for valid triples and low scores for invalid ones, guiding the encoder to capture chemically meaningful hierarchical semantics that reflect underlying reaction transformations. The final product embedding $\mathbf{e}_{\text{product}}$ encodes its structured context and serves as the input

for the retrosynthesis prediction module. Importantly, the architecture is not limited to RGCN and can accommodate alternative KG encoders such as TransE (Bordes et al., 2013), RotatE (Sun et al., 2019), and their variants. Additional details are provided in Appendix B.2.

## 3.2 REACTION-AWARE PRE-TRAINING

To infuse the encoder with retrosynthetic priors, we design a pre-training objective that recovers chemically meaningful fragment sequences from input molecules. Using the BRICS algorithm (Degen et al., 2008), molecules are decomposed by cleaving retrosynthetically significant bonds based on 16 predefined disconnection rules. The resulting fragments retain key substructures and reflect the chemical environment around cleavage sites. Given a molecule $x = [x_1, x_2, \ldots, x_m]$, represented as a SMILES token sequence, we construct the target fragment sequence $u = [u_1, u_2, \ldots, u_L]$ by tokenizing the dot-concatenated fragments. For instance, the molecule *CC(=O)c1ccc2c(ccn2C(=O)OC(C)(C)C)c1* is decomposed into three fragments: *CC=O*, *O=Cn1ccc2cccccc21*, and *CC(C)(C)O*, which are concatenated as *CC=O.O=Cn1ccc2cccccc21.CC(C)(C)O* to form the target. The model is trained autoregressively to generate each fragment token conditioned on the molecule and previously generated tokens:

$$\mathcal{L}_{\text{pretrain}} = -\log P(u|x) = -\sum_{i=1}^{L} \log P(u_i \mid u_{<i}, x). \tag{4}$$

This task enables the encoder to capture fragment-level structural patterns and recognize plausible disconnection sites, providing a stronger initialization for downstream retrosynthesis modeling.

## 3.3 KNOWLEDGE-GUIDED RETROSYNTHESIS PREDICTION

**Task-relevant Knowledge Adapter.** While KG embeddings encode rich structural and relational semantics, they may also contain task-irrelevant or redundant information that hinders downstream performance. Motivated by the variational information bottleneck principle (Sun et al., 2022), we propose a task-relevant knowledge adapter to further filter redundant signals and retain information essential for retrosynthesis. Given a product embedding $\mathbf{e}_{\text{product}} \in \mathbb{R}^d$ obtained from the KG encoder, we model the approximate posterior distribution of the latent variable $\mathbf{z}_{\text{product}}$ as a multivariate Gaussian:

$$p(\mathbf{z}_{\text{product}} \mid \mathbf{e}_{\text{product}}) = \mathcal{N}\left(f_\phi^\mu(\mathbf{e}_{\text{product}}), \; f_\phi^\Sigma(\mathbf{e}_{\text{product}})\right), \tag{5}$$

where $f_\phi^\mu(\cdot)$ and $f_\phi^\Sigma(\cdot)$ denote the neural network outputs for the mean vector and the diagonal covariance matrix, respectively. Specifically, the output of $f_\phi^\Sigma(\cdot)$ is passed through a softplus transformation to ensure positive semi-definiteness, which enables analytical computation of the Kullback–Leibler (KL) divergence (Hershey & Olsen, 2007). A latent vector $\mathbf{z}_{\text{product}}$ is then sampled using the reparameterization trick (Kingma et al., 2013):

$$\mathbf{z}_{\text{product}} = f_\phi^\mu(\mathbf{e}_{\text{product}}) + f_\phi^\Sigma(\mathbf{e}_{\text{product}}) \odot \boldsymbol{\epsilon}, \quad \boldsymbol{\epsilon} \sim \mathcal{N}(\mathbf{0}, \mathbf{I}), \tag{6}$$

where $\odot$ denotes element-wise multiplication. To regularize information flow and retain only task-relevant information, we apply a KL divergence penalty between the approximate posterior and a standard isotropic Gaussian prior, formulated as:

$$\mathcal{L}_{\text{IB}} = D_{\text{KL}}\left(p(\mathbf{z}_{\text{product}} \mid \mathbf{e}_{\text{product}}) \,\|\, \mathcal{N}(\mathbf{0}, \mathbf{I})\right), \tag{7}$$

where $D_{\text{KL}}(\cdot\|\cdot)$ denotes the KL divergence.

**Knowledge Injection.** To incorporate the task-relevant representation into the generation process, $\mathbf{z}_{\text{product}}$ is projected via a learnable linear transformation to obtain $\tilde{\mathbf{e}}_{\text{product}}$, which is then injected into both the encoder and decoder through residual fusion. In the encoder, this is implemented by a residual connection where $\mathbf{h}_{\text{enc}}^{(l+1)} = \text{TransformerEncoder}(\mathbf{h}_{\text{enc}}^{(l)} + \tilde{\mathbf{e}}_{\text{product}})$, where $\mathbf{h}_{\text{enc}}^{(l)}$ denotes the hidden state at layer $l$. In the decoder, $\tilde{\mathbf{e}}_{\text{product}}$ enriches the cross-attention mechanism, with $Q = \mathbf{W}_Q \mathbf{h}_{\text{dec}}^{(l)}$, $K = \mathbf{W}_K(\mathbf{h}_{\text{enc}}^{(l+1)} + \tilde{\mathbf{e}}_{\text{product}})$, $V = \mathbf{W}_V(\mathbf{h}_{\text{enc}}^{(l+1)} + \tilde{\mathbf{e}}_{\text{product}})$. Attention is then computed as $\text{softmax}(QK^\top/\sqrt{d})V$, where $\mathbf{h}_{\text{dec}}^{(l)}$ is the decoder hidden state, and $\mathbf{W}_Q, \mathbf{W}_K, \mathbf{W}_V$ are learnable projection matrices. This dual-path strategy enables more accurate reactant generation.

**Joint Training Objective.** The final training objective jointly minimizes the autoregressive generation loss (Eq. 1) and the KL regularization term (Eq. 7):

$$\mathcal{L}_{\text{total}} = \mathcal{L}_{\text{gen}} + \beta\mathcal{L}_{\text{IB}}, \tag{8}$$

where the coefficient $\beta$ balances generation accuracy against the need to compress information. This objective encourages the model to generate syntactically valid and chemically plausible reactants while incorporating task-relevant structural semantics distilled from the KG.

## 3.4 THEORETICAL ANALYSIS

The knowledge adapter distills task-relevant signals from product embeddings by filtering semantic noise from the knowledge graph, guided by the Information Bottleneck principle, which formalizes the trade-off between informativeness and compression.

**Definition 1 (Information Bottleneck).** *Given a product embedding $\mathbf{e}_{product}$ and its corresponding reactant sequence $y^{rxn}$, the IB objective aims to learn the minimal sufficient representation $\mathbf{z}_{product}$:*

$$\arg\min_{\mathbf{z}_{\text{product}}} -I(\mathbf{z}_{\text{product}}; y^{\text{rxn}}) + \beta I(\mathbf{z}_{\text{product}}; \mathbf{e}_{\text{product}}), \tag{9}$$

*where $I(A; B) = H(A) - H(A|B)$ denotes the Shannon mutual information (Cover, 1999; Ma et al., 2025) and $\beta > 0$ balances task relevance and representation compression.*

Intuitively, the first term encourages alignment with the prediction target, while the second penalizes excessive dependence on the input embedding. To formalize the effect of filtering out irrelevant information, we consider $\mathbf{e}_n$ as a task-independent component of $\mathbf{e}_{\text{product}}$, such as structural noise or KG-specific redundancy. Under the following Markov assumption: $< (y^{\text{rxn}}, \mathbf{e}_n) \rightarrow \mathbf{e}_{\text{product}} \rightarrow \mathbf{z}_{\text{product}} >$, the learned representation $\mathbf{z}_{\text{product}}$ should ideally be invariant to $\mathbf{e}_n$.

**Lemma 1 (Task-Relevant Knowledge Extraction).** *Under the assumption above, the mutual information between $\mathbf{z}_{product}$ and the task-independent component $\mathbf{e}_n$ satisfies:*

$$I(\mathbf{z}_{\text{product}}; \mathbf{e}_n) \leq I(\mathbf{z}_{\text{product}}; \mathbf{e}_{\text{product}}) - I(\mathbf{z}_{\text{product}}; y^{\text{rxn}}). \tag{10}$$

A formal proof is provided in Appendix C.1. Lemma 1 shows that minimizing the IB objective reduces mutual information with irrelevant noise. In practice, we implement this via a KL loss for latent regularization and a generation loss to encourage task-specific representation (see Eq. 8).

## 4 EXPERIMENTS

We investigate the following questions to assess KnowRetro: **RQ1)** Does KnowRetro outperform existing baselines? **RQ2)** How does structured knowledge integration improve retrosynthesis prediction? **RQ3)** Is KnowRetro robust and diverse in leveraging knowledge for retrosynthesis?

### 4.1 EXPERIMENTAL SETUP

We describe the datasets, evaluation metrics, and baselines below. Additional implementation details, including hyperparameters and training procedures, are provided in Appendix A.1 and B.1.

**Datasets.** We utilize two types of data: large-scale unlabeled molecular data, which support both knowledge graph construction and self-supervised pretraining, and benchmark reaction datasets for training and evaluation. *(1) KG construction.* We build a hierarchical chemical KG using molecular structures from USPTO-derived molecules (USPTO-50K (Coley et al., 2017), USPTO-MIT (Jin et al., 2017), USPTO-FULL (Dai et al., 2019; Lowe, 2012)) and 250K additional molecules from ZINC15 (Sterling & Irwin, 2015; Zhang et al., 2021). Each molecule is decomposed into BRICS-based substructures and SMARTS-defined functional groups, and the resulting graph encodes relationships among molecules, substructures, and functional groups. *(2) Pretraining.* For self-supervised pretraining, we use 10M molecules from PubChem (Zeng et al., 2022), where the encoder learns to predict BRICS fragments from input molecules in a reaction-agnostic manner. *(3) Retrosynthesis benchmarks.* We evaluate on two widely used benchmarks. The **USPTO-50K** dataset contains 50,016 reactions across 10 classes with a standard 40K/5K/5K train/validation/test split, evaluated under both *Known* (reaction class provided) and *Unknown* (without reaction class labels)

Table 1: Top-$k$ accuracy (%) on USPTO-50K under reaction class unknown and known settings. Models are grouped by knowledge type. Bold indicates the best result; "–" indicates not reported results or not supported class known setting. Results of USPTO-FULL refer to Appendix B.6.

| Type | Model | Reaction Class Unknown | | | | Reaction Class Known | | | |
|------|-------|-----|-----|-----|-----|-----|-----|-----|-----|
| | | $k=1$ | $k=3$ | $k=5$ | $k=10$ | $k=1$ | $k=3$ | $k=5$ | $k=10$ |
| Rule-based | RetroSim | 37.3 | 54.7 | 63.3 | 74.1 | 52.9 | 73.8 | 81.2 | 88.1 |
| | NeuralSym | 44.4 | 65.3 | 72.4 | 78.9 | 55.3 | 76.0 | 81.4 | 85.1 |
| | GLN | 52.5 | 74.6 | 80.5 | 86.9 | 64.2 | 79.1 | 85.2 | 90.0 |
| | LocalRetro | 53.4 | 77.5 | 85.9 | 92.4 | 63.9 | 86.8 | 92.4 | 96.3 |
| | RetroKNN | **57.2** | **78.9** | **86.4** | **92.7** | **66.7** | **88.2** | **93.6** | **96.6** |
| Knowledge-enhanced | G2Gs | 48.9 | 67.6 | 72.5 | 75.5 | 61.0 | 81.3 | 86.0 | 88.7 |
| | RetroXpert | 50.4 | 61.1 | 62.3 | 63.4 | 62.1 | 75.8 | 78.5 | 80.9 |
| | MEGAN | 48.1 | 70.7 | 78.4 | 86.1 | 60.7 | 82.0 | 87.5 | 91.6 |
| | GraphRetro | 53.7 | 68.3 | 72.2 | 75.5 | 63.9 | 81.5 | 85.2 | 88.1 |
| | Retroformer | 53.2 | 71.1 | 76.6 | 82.1 | 64.0 | 82.5 | 86.7 | 90.2 |
| | G$^2$Retro | 54.1 | 74.1 | 81.2 | 86.7 | 63.1 | 84.2 | 88.5 | 91.7 |
| | PMSR | **62.0** | 78.4 | 82.9 | 86.8 | **67.1** | 82.1 | 88.1 | 92.7 |
| | RetroExplainer | 57.7 | **79.2** | **84.8** | **91.4** | 66.8 | **88.0** | **92.5** | **95.8** |
| Data-driven | SCROP | 43.7 | 60.0 | 65.2 | 68.7 | 59.0 | 74.8 | 78.1 | 81.1 |
| | Aug. Transformer | 53.5 | 69.4 | 81.0 | 85.7 | - | - | - | - |
| | R-SMILES | 56.3 | 79.2 | **86.2** | **91.0** | - | - | - | - |
| | Ualign | 53.5 | 77.3 | 84.6 | 90.5 | **66.4** | **86.7** | **91.5** | **95.0** |
| | EditRetro | **60.8** | **80.6** | 86.0 | 90.3 | - | - | - | - |
| Ours | **KnowRetro** | 62.7 | 82.1 | 88.1 | 92.7 | 71.6 | 90.2 | 93.9 | 96.8 |

settings. We further evaluate KnowRetro on the **USPTO-FULL** dataset, a larger and more diverse benchmark with a broader range of reaction types. Detailed statistics and results are provided in Appendix A.2 and Appendix B.6, respectively.

**Evaluation Metrics.** To effectively assess the performance of models, we employ three metrics: (1) **Top-$k$ Accuracy**, measuring how often the ground truth reactants rank within the Top-$k$ predictions ($k \in \{1, 3, 5, 10\}$) based on canonical SMILES comparison; (2) **MaxFrag Accuracy** (Tetko et al., 2020), inspired by classical retrosynthesis, which assesses the exact match of the largest fragment between predicted and ground truth reactants to address ambiguities in reagent reactions; (3) **Round-Trip Accuracy** (Schwaller et al., 2020), which assesses the chemical validity of predicted reactants by checking if a forward reaction model (Schwaller et al., 2019) can regenerate the original product. These metrics comprehensively assess prediction accuracy, fragment relevance, and chemical validity, with higher values indicating better performance. Details are provided in Appendix A.3.

**Baselines.** (1) **Rule-based methods**, which rely on predefined templates or similarity rules, including RetroSim (Coley et al., 2017), NeuralSym (Segler & Waller, 2017), GLN (Dai et al., 2019), LocalRetro (Chen & Jung, 2021), and RetroKNN (Xie et al., 2023). (2) **Data-driven methods**, which directly learn the mapping of products to reactants through end-to-end architectures, such as SCROP (Zheng et al., 2019), Aug. Transformer (Tetko et al., 2020), R-SMILES (Zhong et al., 2022), Ualign (Zeng et al., 2024), and EditRetro (Han et al., 2024). (3) **Knowledge-enhanced methods**, which incorporate auxiliary supervision from reaction-level annotations (e.g., reaction centers, synthons, or reaction types), represented by RetroXpert (Yan et al., 2020), G2Gs (Shi et al., 2020), MEGAN (Sacha et al., 2021), GraphRetro (Somnath et al., 2021), Retroformer (Wan et al., 2022), G²Retro (Chen et al., 2023), PMSR Jiang et al. (2023), and RetroExplainer (Wang et al., 2023).

### 4.2 COMPARISON WITH BASELINES (RQ1)

To address RQ1, we evaluate the KnowRetro on USPTO-50K, USPTO-FULL (Appendix B.6), different reaction types (Appendix B.3), and zero-shot settings (Appendix B.4).

**Top-$k$ Accuracy.** As shown in Table 1, KnowRetro achieves 62.7% Top-1 accuracy in the class-unknown scenario, surpassing RetroKNN (57.2%) and RetroExplainer (57.7%). These results validate that structural knowledge from unlabeled molecules provides a valuable signal for retrosynthesis and yields performance comparable to or exceeding rule- and knowledge-enhanced methods. In the class-known scenario, KnowRetro reaches 71.6% Top-1 accuracy, representing an improvement of about 4% over the strongest rule- and knowledge-enhanced baselines. Relative to EditRetro, the strongest data-driven baseline, KnowRetro shows consistent improvements, increasing Top-1 accu-

racy by 1.9% and 4.6% in the unknown and known settings, respectively. Paired t-tests confirm that these improvements are statistically significant across all Top-k metrics (e.g., p = 0.0012 for Top-1 and p = 0.0005 for Top-10), underscoring the effectiveness of the proposed framework.

**MaxFrag and Round-Trip Accuracy.** KnowRetro is further evaluated using the MaxFrag metric, which emphasizes chemical reasoning by focusing on primary reactants and addressing ambiguities in reagent reactions (Tetko et al., 2020). It achieves a Top-1 accuracy of 66.9% and a Top-10 accuracy of 94.1% on the USPTO-50K unknown dataset (see Table 2), demonstrating its superior performance in handling diverse and complex reaction scenarios. To further assess the practical utility of KnowRetro, we use the Round-Trip accuracy metric, which eval-

Table 2: Comparison of MaxFrag and Round-Trip Top-$k$ accuracy on USPTO-50K unknown. Baselines follow EditRetro (Han et al., 2024).

| Metric | Model | $k=1$ | $k=3$ | $k=5$ | $k=10$ |
|--------|-------|-------|-------|-------|--------|
| MaxFrag | Aug. Transformer | 58.5 | 73.0 | 85.4 | 90.0 |
| | MEGAN | 54.2 | 75.7 | 83.1 | 89.2 |
| | R-SMILES | 61.0 | 82.5 | 88.5 | 92.8 |
| | EditRetro | 65.3 | 83.9 | 88.9 | 92.8 |
| | **KnowRetro** | **66.9** | **84.5** | **89.9** | **94.1** |
| Round-Trip | Graph2SMILES | 76.7 | 56.0 | 46.4 | 34.9 |
| | Retroformer | 78.9 | 72.0 | 67.1 | 57.2 |
| | EditRetro | **83.4** | 73.6 | 65.3 | 50.8 |
| | **KnowRetro** | 82.2 | **74.9** | **70.5** | **62.0** |

uates whether the predicted reactants can regenerate the target product via a forward reaction model (Schwaller et al., 2019). On the USPTO-50K unknown set, KnowRetro achieves 82.2% Top-1 and 62.0% Top-10 accuracy, outperforming baselines on almost all metrics (Table 2). Notably, while EditRetro achieves the highest Top-1 (83.4%), likely due to self-distillation with forward-validated reactants, KnowRetro achieves more diverse and chemically valid predictions, as reflected in its superior Top-10 Round-Trip accuracy, making it better suited for practical retrosynthesis.

## 4.3 ABLATION STUDY (RQ2)

We perform ablations to assess the impact of each component in **KnowRetro**. Additional experiments on knowledge injection strategies (Appendix B.5) and the knowledge adapter (Appendix B.7) provide further insights into model design and reaction knowledge retention.

**Effect of Pre-training and Hierarchical KG.** (1) *KnowRetro w/o KG & PT* removes both the knowledge graph and pre-training modules, reducing the model to a vanilla Transformer. It achieves a Top-1 accuracy of 56.2%, similar to the Transformer-based R-SMILES (56.3%), showing that KnowRetro without external knowledge performs like a standard sequence model. (2) *KnowRetro w/o KG* adds the pre-training module. It improves the Top-10 accuracy from 90.7% to

Table 3: Ablation results on USPTO-50K (Unknown).

| Model Variant | $k=1$ | $k=3$ | $k=5$ | $k=10$ |
|---------------|-------|-------|-------|--------|
| KnowRetro *w/o KG & PT* | 56.2 | 78.9 | 85.7 | 90.7 |
| KnowRetro *w/o KG* | 57.8 | 80.6 | 86.8 | 92.3 |
| KnowRetro *w/o PT* | 60.4 | 81.3 | 87.3 | 92.2 |
| KnowRetro *w/ Mol* | 56.5 | 80.4 | 86.9 | 92.5 |
| KnowRetro *w/ Mol+Sub* | 60.6 | 81.3 | 86.5 | 92.3 |
| KnowRetro *w/ Mol+Func* | 61.9 | 82.0 | 87.4 | 92.6 |
| **KnowRetro** | **62.7** | **82.1** | **88.1** | **92.7** |

92.3%, indicating that substructure-based pre-training helps the model learn chemical splitting patterns and disconnection strategies. (3) *KnowRetro w/o PT* keeps the hierarchical KG but removes pre-training. The Top-1 accuracy increases from 56.2% to 60.4%, showing that the KG provides useful chemical context through hierarchical structured relationships.

**Effect of Hierarchical Knowledge Structure.** To further examine how each level of structural knowledge contributes to retrosynthesis, we compare four KG variants. Using only molecule-level information (*Mol*), constructed from Morgan fingerprint similarity (Tanimoto > 0.5), results in the lowest Top-1 accuracy (56.5%), suggesting that coarse molecular similarity offers limited additional guidance for retrosynthesis. Using only substructures (*Mol+Sub*) or only functional groups (*Mol+Func*) leads to lower Top-1 accuracy (60.6% and 61.9%, respectively), indicating that each captures only part of the retrosynthetic context. The full KG (*Mol+Sub+Func*) achieves the best result (62.7%), highlighting the complementary roles of substructures in capturing disconnection patterns and functional groups in modeling reactivity. This result demonstrates that our hierarchical KG, learned from large-scale unlabeled molecular data, provides complementary multi-level chemical signals and offers clear benefits for guiding retrosynthesis.

## 4.4 FURTHER STUDY OF CHEMICAL KNOWLEDGE (RQ3)

**Reliability of KnowRetro on Noisy KGs.** KGs in practice may suffer from two types of perturbations: (i) **structural perturbation**, such as missing edges due to incomplete coverage, and (ii) **semantic perturbation**, such as mislabeled relations or functional group misannotations. To assess robustness, we simulate structural perturbation by randomly removing 15%, 25%, and 35% of edges, and semantic perturbation by randomly corrupting relation types or entities. As shown in Figure 3, KnowRetro remains stable across both scenarios. For structural perturbation (Figure 3a), Top-1 accuracy decreases from 62.7% to 58.6% and Top-10 from 92.7% to 88.7%, corresponding to only a 6.5% relative drop in Top-1 accuracy under 35% edge removal. For semantic perturbation (Figure 3b), KnowRetro remains robust, with merely a 2.5% decline in Top-1 accuracy under 35% corruption. These results demonstrate that KnowRetro is resilient to both incomplete and erroneous knowledge graphs, ensuring reliability in noisy real-world scenarios.

**Diversity of Predicted Reactions.** We assess the structural diversity of predicted reactants using multiple molecular similarity metrics, including Tanimoto similarity with Morgan fingerprints, MACCS keys, and Bemis–Murcko scaffolds. Following prior work (Han et al., 2024), we first compute Tanimoto similarity (Bajusz et al., 2015) with 2048-bit Morgan fingerprints, where lower values indicate higher diversity (details in Appendix A.4). As shown in Figure 4, 52.2% of test products fall into high-diversity groups (0.38–0.49), 36.6% into medium (0.53–0.63), and only

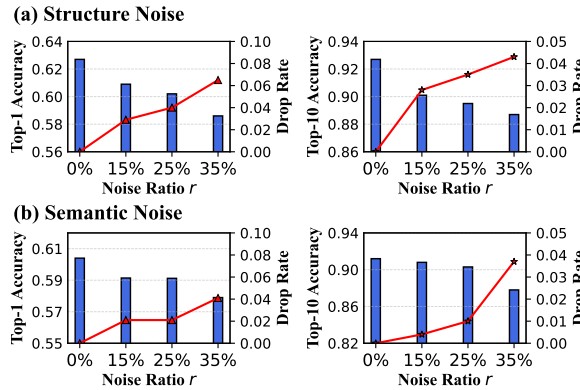

Figure 3: Robustness of **KnowRetro** on USPTO-50K under (a) structural and (b) semantic noise levels. Blue bars show Top-$k$ accuracy; red lines indicate degradation, with smaller drops reflecting greater robustness.

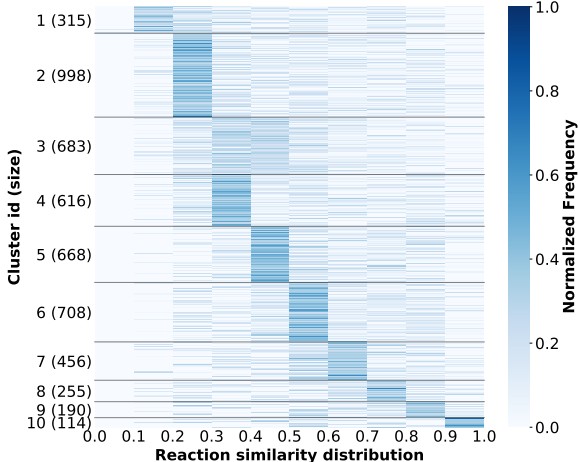

Figure 4: Cluster analysis of test product similarity distributions. Rows indicate clusters; columns denote similarity bins (0.0–1.0). Darker shades represent higher normalized frequencies.

11.2% into low ($\geq$0.65), showing that KnowRetro generates diverse reactants for most products. Compared with EditRetro, KnowRetro achieves lower average similarity (0.49 vs. 0.55), and results from MACCS keys and Bemis–Murcko scaffolds (Appendix B.10) further support its ability to generate diverse reactants while maintaining high prediction accuracy.

## 4.5 CASE STUDY

**Attention Analysis with Hierarchical Chemical Knowledge.** To better illustrate how KnowRetro leverages structural knowledge, we compare the attention behaviors of KnowRetro and a variant without KG (Figure 5). The variant model shows diffuse patterns that mainly follow SMILES syntax and fail to emphasize the true reaction center. In contrast, KnowRetro consistently highlights the chemically meaningful disconnection site, such as the O–C cleavage in this example, and the fragment directly attached to it (e.g. the benzyl group). This joint attention on the disconnection site and its neighboring substructure indicates that the model captures transformation-relevant chemical context rather than relying on SMILES-pattern patterns. Such behavior demonstrates that the hierarchical knowledge learned from large-scale molecular data enables KnowRetro to make more

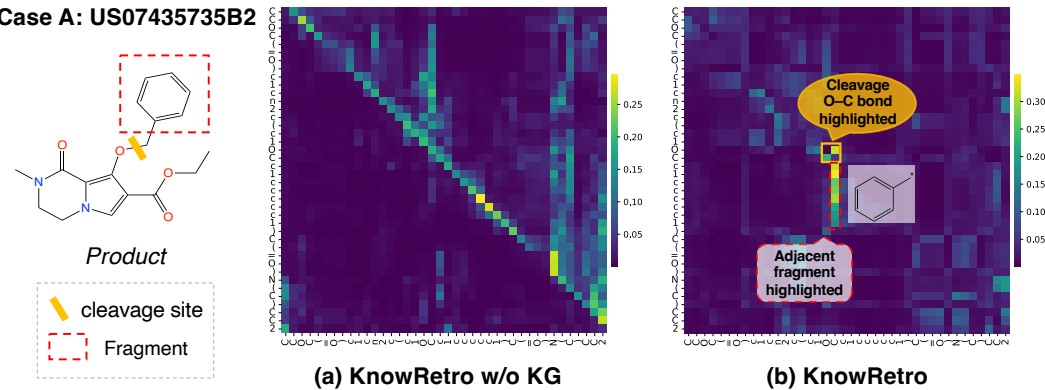

Figure 5: Attention patterns with and without hierarchical knowledge.

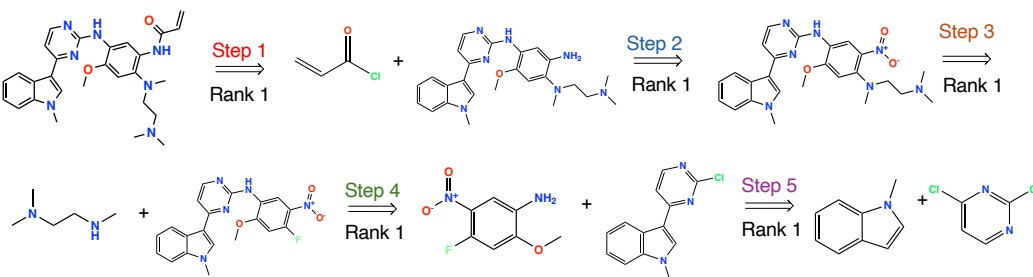

Figure 6: Multistep retrosynthesis generated by KnowRetro for Osimertinib (DB09330).

accurate and chemically grounded disconnection decisions. Additional examples showing consistent behavior across different reaction types are provided in Appendix B.11.

**Multistep Retrosynthesis Planning.** To further evaluate the practical applicability of KnowRetro in real synthesis scenarios, we examine its multistep planning performance by iteratively applying the single-step model at each retrosynthetic stage. Figure 6 illustrates the retrosynthetic route generated by KnowRetro for Osimertinib (DB09330) (Finlay et al., 2014), a third-generation EGFR inhibitor widely used as a targeted anticancer drug. KnowRetro successfully recovers a chemically reasonable five-step route in which each predicted step closely matches known synthetic transformations. This demonstrates that the hierarchical knowledge learned from large-scale molecular data supports coherent, mechanism-consistent disconnections that extend naturally to multistep synthesis. A further example on Lenalidomide (DB00480) (Ponomaryov et al., 2015) is included in Appendix B.12.

## 5  CONCLUSION

We presented **KnowRetro**, a knowledge-guided framework that leverages large-scale unlabeled molecular data for retrosynthesis prediction. By constructing a hierarchical chemical knowledge graph and integrating it through guided pre-training and knowledge-adaptive fine-tuning, KnowRetro captures transferable reaction patterns beyond annotated datasets. Extensive experiments demonstrate consistent gains in accuracy, robustness, and diversity over strong baselines. Beyond retrosynthesis, our work illustrates how structured knowledge from unlabeled data can be systematically incorporated into neural generation, suggesting a general paradigm for improving learning in data-scarce scientific domains. Future work will focus on enriching KnowRetro with quantum-chemical knowledge and experimental data, providing a more comprehensive foundation for knowledge-guided retrosynthesis.

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

# APPENDIX

## A  IMPLEMENTATION DETAILS

### A.1  HYPER-PARAMETER SETTINGS

The KnowRetro framework combines a relational graph encoder with a Transformer-based generation module to jointly model hierarchical chemical knowledge and retrosynthetic sequence prediction. The graph encoder employs a 2-layer RGCN with 256-dimensional node embeddings to encode multi-relational chemical semantics. To improve scalability, we adopt global edge sampling with 30,000 edges per subgraph and apply dropout with a rate of 0.2. Negative samples are constructed by randomly corrupting either the head or tail entity in knowledge triples at a 16:1 ratio. Parameters are initialized using a uniform distribution $U\left[-\frac{6}{\sqrt{d}}, \frac{6}{\sqrt{d}}\right]$, with $d = 256$. Gradient clipping with a maximum norm of 1.0 is applied to stabilize training. The sequence generator is an 8-layer Transformer with 256 hidden dimensions and 8 attention heads in both encoder and decoder. Optimization uses Adam with a Noam learning rate schedule (8000 warm-up steps) and 0.3 dropout. The variational bottleneck parameter is set to $\beta = 0.03$, which provides the best trade-off between accuracy and regularization (see Appendix B.8 for a sensitivity analysis). Pre-training is conducted for 2 million steps with a batch size of 128, followed by fine-tuning for 400,000 steps with a batch size of 64. All experiments were repeated 10 times with different random seeds to ensure statistical robustness. The USPTO-50K dataset is augmented 20× following (Zhong et al., 2022), and the model is trained for 24 hours on a single NVIDIA GeForce RTX 3090 GPU with 24GB memory. See AppendixB.1 for augmentation details.

### A.2  DATASETS

We evaluate on two USPTO-derived benchmarks. The **USPTO-50K** dataset (Coley et al., 2017) contains 50,016 atom-mapped reactions across 10 classes with a standard 40K/5K/5K train/validation/test split, and is evaluated under both *Known* (reaction class provided) and *Unknown* (without reaction class labels) settings. The **USPTO-FULL** dataset (Dai et al., 2019; Lowe, 2012) consists of 950,000 cleaned reactions extracted

Table 4: Statistics of node in the KG.

| Node Type | Count |
|---|---|
| Molecule | 1,353,930 |
| Substructure | 414,640 |
| Functional Group | 964 |

from U.S. patents (1976–2016), where multi-product reactions are separated into single-product entries and the data is split into 80%/10%/10% for training, validation, and testing. For knowledge graph construction, we use only *molecular structures* from USPTO-50K, USPTO-MIT, and USPTO-FULL, together with 250K additional molecules sampled from ZINC15 (Sterling & Irwin, 2015; Zhang et al., 2021). Each molecule is decomposed into BRICS-based substructures and SMARTS-defined functional groups, while reaction-level information (e.g., reactant–product pairs or labels) is not used. All molecules are canonicalized and deduplicated prior to graph construction. The resulting hierarchical KG encodes relationships among molecules, substructures, and functional groups, and contains 1,769,534 entities and 15,406,980 triples connected by seven relation types. Table 4 reports the node statistics. **The reported performance of baselines is from the original papers for comparison**.

### A.3  DETAILED EXPLANATION OF EVALUATION METRICS

To comprehensively evaluate the performance of KnowRetro, we introduce three key metrics: Top-$k$ Accuracy, MaxFrag Accuracy, and Round-Trip Accuracy. The details are as follows:

**Top-$k$ Accuracy.** Top-$k$ Accuracy evaluates the ability of the model to rank the ground truth reactants among its top-$k$ predictions. The metric is defined as:

$$\text{Top-}k \text{ Accuracy} = \frac{1}{N}\sum_{i=1}^{N}\mathbb{I}\left[Y_{\text{true}} \in \{Y_i^j \mid j = 1, \ldots, k\}\right],$$

where $N$ is the total number of test products, $Y_{\text{true}}$ represents the ground truth reactants for the $i$-th product, $Y_i^j$ denotes the $j$-th predicted reactant set for the $i$-th product, and $\mathbb{I}[\cdot]$ is the indicator function that returns 1 if the ground truth reactants appear in the top-$k$ predictions, and 0 otherwise.

This metric evaluates the model's recall, focusing on its ability to retrieve the correct reactants from a limited set of predictions.

**MaxFrag Accuracy.** MaxFrag Accuracy, inspired by classical retrosynthesis (Tetko et al., 2020), evaluates the exact match of the largest reactant fragment between predicted and ground truth reactants, specifically addressing ambiguities in reagent selection. The calculation involves canonicalizing the reactant SMILES strings, removing atom mapping numbers, and splitting the SMILES string by the dot (.) to identify individual fragments. The largest fragment is determined based on atom count. The metric is defined as:

$$\text{MaxFrag}(k) = \frac{1}{N} \sum_{i=1}^{N} \mathbb{I} \left[ f(Y_{\text{true}}) \in \{ f(Y_i^j) \mid j = 1, \ldots, k \} \right],$$

where $N$ denotes the total number of test cases, $Y_{\text{true}}$ represents the ground truth reactants for the $i$-th product, $Y_i^j$ refers to the $j$-th predicted reactant set for the $i$-th product, $f(\cdot)$ is a function extracting the largest fragment by atom count, and $\mathbb{I}[\cdot]$ is an indicator function that returns 1 if the largest fragment of the ground truth reactants matches any of the top-$k$ predicted reactants and 0 otherwise. This metric focuses on the primary reactive component, ensuring a robust evaluation of retrosynthesis performance, particularly under scenarios with ambiguous reagents.

**Round-Trip Accuracy.** Round-Trip Accuracy(Schwaller et al., 2020) measures the feasibility of predicted reactants by verifying whether the original product can be regenerated using a forward reaction model (e.g., Molecular Transformer (Schwaller et al., 2019)). For Top-$k$ predictions, the metric is computed as:

$$\text{RoundTrip}(k) = \frac{1}{N \cdot k} \sum_{i=1}^{N} \sum_{j=1}^{k} \mathbb{I} \left[ \text{Match Product} \right],$$

where $N$ is the total number of test products, $k$ represents the number of top predictions considered, and $\mathbb{I}[\cdot]$ is an indicator function that returns 1 if the forward reaction model successfully reproduces the original product from the predicted reactants, and 0 otherwise. By emphasizing chemical validity and practical feasibility, this metric complements Top-$k$ Accuracy by providing an additional evaluation criterion for retrosynthesis predictions.

### A.4 Details on reaction diversity analysis

To assess the structural diversity of predicted reactions, we calculated pairwise Tanimoto similarities among the top-10 predicted reactants for each product and clustered the products based on their reaction similarity distributions (Chen et al., 2023). Algorithm 1 outlines the workflow, including similarity computation and clustering. The pairwise reaction similarities, as detailed in the fourth line of the algorithm, are computed in the following scenarios: (1) **One-to-One**: When both reactant sets contain a single reactant, the Tanimoto similarity (Bajusz et al., 2015) is calculated by comparing their molecular fingerprints. The similarity measures the overlap between the two sets of fingerprints. (2) **One-to-Multiple**: If one reactant set contains a single reactant and the other contains multiple reactants, the molecular fingerprints of the multi-reactant set are combined using a bitwise OR operation. The Tanimoto similarity is then computed between the merged fingerprint of the multi-reactant set and the fingerprint of the single reactant. (3) **Multiple-to-Multiple**: For scenarios where both reactant sets contain multiple reactants, we focus on cases with exactly two reactants in each set. The pairwise similarity is calculated by comparing each reactant in one set to each reactant in the other set, then averaging the highest similarity values.

## B Additional Experiments

### B.1 Effect of Data augmentation

We follow the data augmentation strategy proposed in previous works (Zhong et al., 2022), where augmentation is applied both during training and testing. Specifically, we perform 20× data augmentation for both the training and test sets of the USPTO-50K dataset. During training, we generate multiple input-output pairs by enumerating different atoms as the root of the SMILES string, thereby

---

**Algorithm 1** Workflow for Reaction Diversity Calculation

---

**Input**: Set of $K$ products with their top-10 predicted reactant sets $\{X^k, \{Y_i^k\}_{i=1,\ldots,10}\}_{k=1,\ldots,K}$, and the number of clusters $N_{\text{Clusters}}$.
**Output**: Clusters of products $\{C_i\}_{i=1,\ldots,N_{\text{Clusters}}}$ based on reaction similarity distributions.

1: **for** $k = 1$ to $K$ **do**
2:     Initialize the set of pairwise similarities $\{\text{sim}_{ij}^k\}$ for product $X^k$.
3:     **for** each pair of reactions $(Y_i^k, Y_j^k)$ where $i \neq j$ **do**
4:         Compute similarity: $\text{sim}_{ij}^k = \text{Tanimoto}(Y_i^k, Y_j^k)$.
5:     **end for**
6:     Generate the reaction similarity distribution for product $X^k$ as a histogram: $h^k = \text{Histogram}(\{\text{sim}_{ij}^k\})$.
7:     Normalize the histogram: $\hat{h}^k = h^k / \sum h^k$.
8: **end for**
9: Apply K-Means clustering to the set of normalized histograms $\{\hat{h}^k\}_{k=1,\ldots,K}$: $\{C_i\} = $ K-Means$(\{\hat{h}^k\}, N_{\text{Clusters}})$
10: **for** $i = 1$ to $N_{\text{Clusters}}$ **do**
11:     Compute the average reaction similarity for cluster $C_i$: $\overline{\text{sim}}_i = \frac{1}{|C_i|} \sum_{k \in C_i} \text{Average}(\{\text{sim}_{ij}^k\})$.
12: **end for**
13: Sort clusters $\{C_i\}$ in ascending order of average similarity $\overline{\text{sim}}_i$ (lower $\overline{\text{sim}}_i$ indicates higher diversity).
14: **return** $\{C_i\}_{i=1,\ldots,N_{\text{Clusters}}}$ (Clustered products and their similarity distributions)

---

increasing the diversity of training examples and improving model generalization. During inference, multiple SMILES variants of the same molecule are used to generate diverse predictions, which are uniformly scored and aggregated by top-K selection.

Applying $20\times$ training augmentation alone already improves KnowRetro's Top-1 accuracy from 57.1% to 60.4%, , while R-SMILES improves from 40.9% to 51.2% under the same setting. We then further evaluated the effect of test-time augmentation, as presented in Table 5. For R-SMILES, Top-1 accuracy improved from 51.2% to 56.3%, and Top-10 accuracy increased from 83.0% to 91.0%. KnowRetro exhibited similar gains, with Top-1 accuracy rising from 60.4% to 62.7%, and Top-10 accuracy

Table 5: Comparison of Top-$k$ performance with and without data augmentation during testing. * indicates augmented versions.

| Model | $k=1$ | $k=3$ | $k=5$ | $k=10$ |
|---|---|---|---|---|
| R-SMILES | 51.2 | 75.1 | 81.1 | 83.0 |
| R-SMILES* | 56.3 | 79.2 | 86.2 | 91.0 |
| KnowRetro | 60.4 | 80.5 | 85.7 | 91.2 |
| **KnowRetro*** | **62.7** | **82.1** | **88.1** | **92.7** |

increasing from 91.2% to 92.7% with augmentation (**KnowRetro***). These results demonstrate that data augmentation consistently enhances performance. Notably, even without any test augmentation, the models still performed robustly, with KnowRetro achieving 60.4% Top-1 and 91.2% Top-10 accuracy. The model demonstrates strong performance and adaptability in multi-step inference, with data augmentation offering significant improvements while maintaining strong accuracy without it.

### B.2 PERFORMANCE EVALUATION OF DIFFERENT KG ALGORITHMS

To further validate the effectiveness of incorporating hierarchical chemical knowledge into retrosynthesis, we conducted experiments by integrating several commonly used knowledge graph representation methods into our framework. Specifically, we equip KnowRetro with TransE (Bordes et al., 2013), RotatE (Sun et al., 2019), and RGCN (Schlichtkrull et al., 2018) to evaluate whether the integration of diverse KG

Table 6: Top-$k$ accuracy under different KG designs.

| Model | $k=1$ | $k=3$ | $k=5$ | $k=10$ |
|---|---|---|---|---|
| R-SMILES | 56.3 | 79.2 | 86.2 | 91.0 |
| KnowRetro-*TransE* | 61.7 | 80.0 | 86.9 | 91.3 |
| KnowRetro-*RotatE* | **64.5** | **82.4** | 88.0 | 92.2 |
| KnowRetro-*RGCN* | 62.7 | 82.1 | **88.1** | **92.7** |

techniques could consistently enhance performance. The results, presented in Table 6, demonstrate

that all KG-based methods significantly outperform the baseline R-SMILES across various Top-$k$ metrics. Notably, KnowRetro-RGCN achieves the highest Top-5 and Top-10 accuracies (88.1% and 92.7%, respectively), while KnowRetro-RotatE achieves the best Top-1 performance at 64.5%. These consistent improvements across different KG representation methods demonstrate the effectiveness and robustness of our knowledge-guided approach. Notably, although RotatE performs slightly better on Top-1, we adopt RGCN as the default encoder due to its inductive nature and stronger generalization to unseen molecular entities.

### B.3 PERFORMANCE ACROSS DIFFERENT REACTION TYPES

To evaluate the model's performance across different reaction types, we compared various variants of KnowRetro, including RGCN, TransE, and RotatE, with R-SMILES (Zhong et al., 2022) and G²Retro (Chen et al., 2023). R-SMILES learns reaction transformations by analyzing input-output SMILES similarity, while G²Retro uses reaction center prediction to model chemical rules. As shown in Figure 7, the RGCN, TransE, and RotatE variants of KnowRetro outperform R-SMILES and G²Retro in 7 out of 10 reaction types, with one type achieving identical performance. This demonstrates the advantage of integrating hierarchical chemical knowledge into the model. By leveraging structured external knowledge,

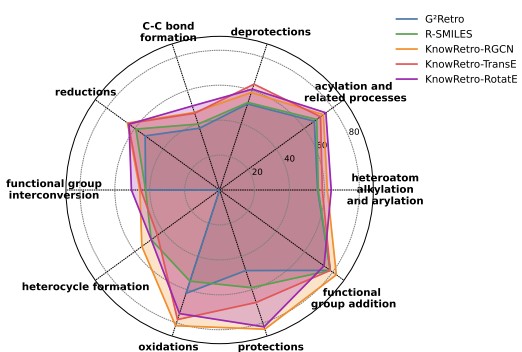

Figure 7: Top-1 accuracy of KnowRetro (and variants) vs. baselines across reaction types.

KnowRetro enhances its adaptability and accuracy across a wide range of reaction types.

### B.4 ZERO-SHOT PERFORMANCE ON FUNCTIONAL GROUP INTERCONVERSION

To assess the knowledge graph's ability to generalize to unseen reaction types, we conducted a zero-shot experiment by removing the Functional Group Interconversion reaction class (type 9) from the training and validation sets, leaving it in the test set. This setup allows evaluation of the KG's role in capturing unseen reaction classes. As shown in Table 7, KnowRetro, which integrates the KG, achieves a significantly higher Top-1 accuracy of 9.4% compared to models without the KG, such as

Table 7: Zero-shot performance on Functional Group Interconversion (Type 9), comparing Top-$k$ accuracy (no test augmentation) with and without KG utilization.

| Model | $k=1$ | $k=3$ | $k=5$ | $k=10$ |
|---|---|---|---|---|
| KnowRetro w/o KG & PT | 5.9 | 7.6 | 9.7 | 12.5 |
| KnowRetro w/o KG | 6.5 | 9.8 | 10.3 | 12.5 |
| KnowRetro w/o PT | 7.8 | 13.2 | 16.4 | 20.2 |
| **KnowRetro** | **9.4** | **14.2** | **16.8** | **20.8** |

KnowRetro w/o KG (6.5%) and KnowRetro w/o KG & PT (5.9%). Similarly, in Top-10 accuracy, KnowRetro achieves 20.8%, outperforming KnowRetro w/o KG (12.5%) and KnowRetro w/o KG & PT (12.5%). Furthermore, when compared to RetroKNN (Xie et al., 2023), as presented in Figures 4a and 4c of the RetroKNN paper, RetroKNN shows comparatively lower performance on type 9 in zero-shot scenarios, with Top-5 accuracy around 8.5% and Top-10 accuracy around 12.5%. These results highlight the critical role of the KG in enabling KnowRetro to capture hierarchical chemical relationships, thereby enhancing its performance in zero-shot scenarios.

### B.5 EFFECT OF KNOWLEDGE INJECTION STRATEGIES

We study the impact of different knowledge injection strategies in KnowRetro by comparing three configurations: (i) encoder-only injection, (ii) decoder-only injection, and (iii) combined encoder–decoder injection. These strategies determine where the distilled KG embedding $z_{product}$ is fused into the generative model.

Table 8: Effect of knowledge injection strategies.

| Injection Strategy | $k=1$ | $k=3$ | $k=5$ | $k=10$ |
|---|---|---|---|---|
| Encoder-only | 61.4 | 81.6 | 87.5 | 92.4 |
| Decoder-only | 60.9 | 81.4 | 87.0 | 92.2 |
| **Enc–Dec (Ours)** | **62.7** | **82.1** | **88.1** | **92.7** |

As shown in Table 8, injecting knowledge into the encoder yields better results than decoder-only injection, indicating that early conditioning helps the model better understand chemical structures. Decoder-only injection also improves accuracy by guiding generation with reaction-aware context. The encoder–decoder configuration achieves the best performance, confirming that injecting knowledge into both stages allows the model to more effectively utilize hierarchical chemical information. These results highlight the complementary roles of encoder and decoder injection, demonstrating that joint conditioning at both levels is essential for improving retrosynthesis prediction.

## B.6 Performance on the USPTO-FULL dataset

To evaluate the scalability and generalization of KnowRetro, we conducted experiments on the USPTO-FULL dataset (Dai et al., 2019), a benchmark significantly larger and more diverse than USPTO-50K. This dataset contains approximately 950,000 reactions spanning a wide range of reaction types, with 80% allocated for training, 10% for validation, and 10% for testing, making it an ideal benchmark for assessing the robustness of KnowRetro across diverse reaction datasets. To ensure fair comparison, KnowRetro is trained on a $5\times$ augmented USPTO-FULL dataset, following the setup of R-SMILES (Zhong et al., 2022) and EditRetro (Han et al., 2024), while all other baselines use their original settings. As shown in Table 9, KnowRetro achieves the best overall performance, with 56.2% Top-1 and 80.1% Top-10 accuracy. These results highlight its clear advantage over existing rule-based, task-guided, and data-driven methods, demonstrating

Table 9: Top-$k$ accuracy (%) on USPTO-FULL dataset. "–" denotes results not reported.

| Model | $k{=}1$ | $k{=}3$ | $k{=}5$ | $k{=}10$ |
|---|---|---|---|---|
| RetroSim | 32.8 | – | – | 56.1 |
| NeuralSym | 35.8 | – | – | 60.8 |
| GLN | 39.3 | – | – | 63.7 |
| LocalRetro | 39.1 | 53.3 | 58.4 | 63.7 |
| RetroXpert | 49.4 | 63.6 | 67.6 | 71.6 |
| RetroPrime | 44.1 | – | – | 68.5 |
| RetroExplainer | 51.4 | 70.7 | 74.7 | 79.2 |
| Aug. Transformer | 46.2 | – | – | 73.3 |
| MEGAN | 33.6 | – | – | 63.9 |
| Graph2Edits | 44.0 | 60.9 | 66.8 | 72.5 |
| R-SMILES | 48.9 | 66.6 | 72.0 | 76.4 |
| PMSR | 45.5 | 60.9 | 65.5 | 70.1 |
| Ualign | 50.4 | 66.1 | 71.3 | 76.2 |
| EditRetro | 52.2 | 67.1 | 71.6 | 74.2 |
| **KnowRetro** | **56.2** | **71.4** | **75.7** | **80.1** |

that hierarchical knowledge extracted from unlabeled molecules enables robust generalization across large-scale and diverse reaction spaces.

Beyond Top-$k$ accuracy, we further evaluate KnowRetro on USPTO-FULL using MaxFrag and Round-Trip metrics. As shown in Table 10, KnowRetro consistently surpasses EditRetro, achieving higher MaxFrag accuracy, which reflects more reliable alignment with major product fragments, and substantially higher Round-Trip accuracy, indicating chemically valid and reversible predictions. In addition, diversity analysis (Table 11) shows that KnowRetro generates a markedly larger proportion of high-diversity predictions, underscoring its stronger capacity to explore alternative disconnection strategies. These results demonstrate that KnowRetro generalizes robustly to large and diverse benchmarks while delivering accurate and diverse predictions.

Table 10: Comparison on USPTO-FULL with respect to MaxFrag and RoundTrip accuracy.

| Metrics | MaxFrag | | RoundTrip | |
|---|---|---|---|---|
| | EditRetro | KnowRetro | EditRetro | KnowRetro |
| Top-1 | 59.9 | **64.6** | 75.7 | **76.1** |
| Top-3 | 72.2 | **78.0** | 55.1 | **64.3** |
| Top-5 | 75.3 | **81.7** | 43.8 | **58.1** |
| Top-10 | 78.2 | **85.4** | 31.0 | **50.0** |

Table 11: Diversity on USPTO-FULL.

| Diversity Level | EditRetro (%) | KnowRetro (%) |
|---|---|---|
| High Diversity | 33.0 | 51.0 |
| Medium Diversity | 45.0 | 20.5 |
| Low Diversity | 22.0 | 28.5 |

## B.7 Effectiveness of the Task-Relevant Knowledge Adapter

To evaluate the contribution of the task-relevant knowledge adapter, we conducted a controlled experiment by removing this module and directly injecting the raw KG embeddings into the encoder and decoder (denoted as KnowRetro w/o Adapter). As shown in Table 12, removing the Knowledge Adapter results in consistent

Table 12: Impact of the Knowledge Adapter on USPTO-50K (Unknown).

| Model Variant | $k{=}1$ | $k{=}3$ | $k{=}5$ | $k{=}10$ |
|---|---|---|---|---|
| KnowRetro w/o Adapter | 61.5 | 81.4 | 87.5 | 92.2 |
| **KnowRetro** | **62.7** | **82.1** | **88.1** | **92.7** |

performance drops across all Top-$k$ metrics, with Top-1 accuracy declining by 1.2%. These results underscore the importance of the adapter in suppressing irrelevant signals and enhancing the utility of KG-based features for accurate prediction.

To further probe the role of the adapter, we visualize reaction embeddings before and after its application using UMAP (Figure 8). Before passing through the adapter, embeddings of different reaction classes show substantial overlap. After integration, four major reaction categories, namely heteroatom alkylation and arylation, acylation, C–C bond formation, and deprotections, each accounting for more than 10% of the test dataset, form compact and well-separated clusters. These results demonstrate that the adapter enhances both intra-class consistency and inter-class separability in the

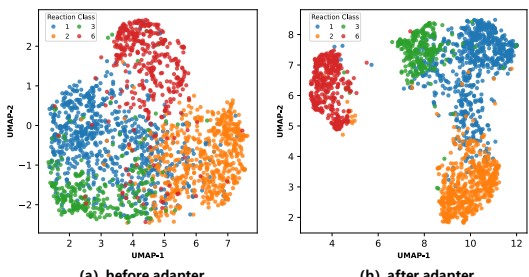

(a) before adapter     (b) after adapter

Figure 8: UMAP visualization of reaction embeddings before and after adapter.

learned representations, even without explicit reaction type supervision, highlighting its role in injecting task-relevant chemical knowledge into retrosynthesis modeling.

## B.8    ANALYSIS OF THE VARIATIONAL BOTTLENECK PARAMETER $\beta$

To assess the impact of the variational bottleneck, we conduct a controlled study on the $\beta$ parameter in Eq. (8) of the main paper, which regulates the strength of the KL divergence regularization. As shown in Table 13, performance remains broadly stable as $\beta$ varies from 0.03 to 1.0, with the best Top-1 accuracy observed at $\beta = 0.03$. These results suggest that moderate values of $\beta$ strike an effective balance between prediction accuracy and regularization strength.

Table 13: Effect of the $\beta$ on USPTO-50K.

| $\beta$ | Top-1 | Top-3 | Top-5 | Top-10 |
|---|---|---|---|---|
| 0.03 | **60.36** | **80.49** | **85.74** | **91.23** |
| 0.1 | 60.25 | 79.85 | 84.42 | 89.49 |
| 0.3 | 60.14 | 79.81 | 85.12 | 90.55 |
| 0.5 | 59.54 | 79.95 | 85.46 | 90.55 |
| 1.0 | 58.48 | 79.91 | 85.54 | 90.85 |

## B.9    ANALYSIS OF THE IMPACT OF THE PRE-TRAINING STRATEGY

We evaluate the effect of the pre-training strategy on the performance of the KnowRetro model (no test-time augmentation). Specifically, we compare KnowRetro, which includes a pre-training phase, with KnowRetro w/o PT, which removes it. As shown in Figure 9, KnowRetro (blue line) consistently outperforms KnowRetro w/o PT (red line) in terms of Top-1 accuracy throughout the training process. Notably, KnowRetro achieves higher accuracy from the early stages of training and maintains superior performance over time. In contrast, KnowRetro w/o PT shows slower improvements, particularly in the early training stages. These findings highlight the essential role of pre-training in improving both the learning efficiency and effectiveness of the model.

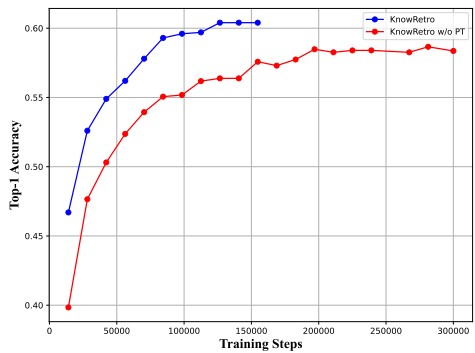

Figure 9: Top-1 accuracy comparison between KnowRetro and KnowRetro w/o PT.

## B.10    ADDITIONAL ANALYSES OF REACTION DIVERSITY

In addition to the Morgan fingerprint–based clustering analysis, we further evaluate diversity using MACCS keys and Bemis–Murcko scaffolds, which capture complementary aspects of molecular variation. MACCS keys represent a set of predefined structural fragments and quantify substructure-level diversity. We compute pairwise similarities among the top-10 predictions for each product and

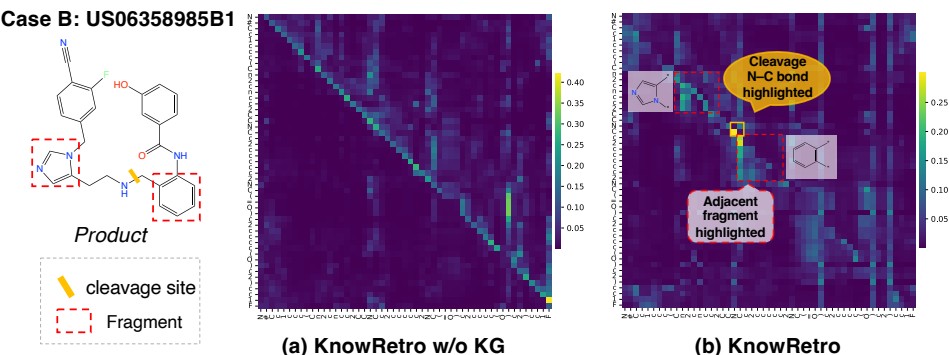

Figure 10: Attention patterns with and without hierarchical knowledge.

**Lenalidomide (DB00480)**

Figure 11: Multistep retrosynthesis generated by KnowRetro for Lenalidomide (DB00480).

report the proportion of pairs below a threshold of 0.7. KnowRetro achieves 22.7% of prediction pairs below this threshold, compared with 20.0% for EditRetro, indicating that our model generates more chemically diverse reactants at the substructure level. Bemis–Murcko scaffolds, in contrast, reflect the core molecular frameworks and provide a measure of scaffold-level diversity. KnowRetro produces an average of 3.59 unique scaffolds per product, with 90.2% and 48.2% of products containing at least 2 and 4 distinct scaffolds, respectively. This level of scaffold diversity is comparable to methods such as EditRetro (4.04 on average), which achieve diversity through multi-site editing around predicted reaction centers. These complementary analyses further confirm that KnowRetro maintains substantial diversity across both substructure and scaffold levels, underscoring its applicability in broad retrosynthetic exploration.

### B.11 ATTENTION ANALYSIS WITH HIERARCHICAL KNOWLEDGE.

Figure 10 shows that the model without KG produces diffuse, syntax-driven attention and fails to highlight the true cleavage site. KnowRetro, however, concentrates on both the reaction center (e.g. N–C bond) and the fragments directly attached to it. This localized and chemically consistent pattern indicates that hierarchical knowledge enables the model to focus on transformation-relevant structures, supporting more accurate and coherent retrosynthetic predictions.

### B.12 ADDITIONAL CASE OF MULTISTEP PLANNING

To further validate the robustness of KnowRetro in multistep synthesis planning, we present an additional example on Lenalidomide (DB00480) (Ponomaryov et al., 2015), an immunomodulatory drug widely used in the treatment of multiple myeloma. Figure 11 shows the retrosynthetic route

produced by iteratively applying the single-step model. KnowRetro produces a concise three-step route that aligns with known transformations, demonstrating coherent disconnection choices across steps and further supporting the effectiveness of the hierarchical knowledge in multistep reasoning.

## C   PROOF

### C.1   PROOF OF LEMMA 1 (*Task-relevant Knowledge Extraction*)

We provide the proof of Lemma 1.

*Proof.* We prove Lemma 1 following the strategy of Proposition 3.1 in (Achille & Soatto, 2018). Suppose the product embedding $\mathbf{e}_{\text{product}}$ is determined by the target $y^{\text{rxn}}$ and task-irrelevant noise $\mathbf{e}_n$, and the latent representation $\mathbf{z}_{\text{product}}$ depends on $\mathbf{e}_n$ only through $\mathbf{e}_{\text{product}}$. We define the Markov Chain $< (y^{rxn}, \mathbf{e}_n) \to \mathbf{e}_{\text{product}} \to \mathbf{z}_{\text{product}} >$. According to the data processing inequality (DPI), it follows that:

$$\begin{aligned}
I(\mathbf{z}_{\text{product}}; \mathbf{e}_{\text{product}}) &\geq I(\mathbf{z}_{\text{product}}; y^{rxn}, \mathbf{e}_n) \\
&= I(\mathbf{z}_{\text{product}}; \mathbf{e}_n) + I(\mathbf{z}_{\text{product}}; y^{rxn}|\mathbf{e}_n) \\
&= I(\mathbf{z}_{\text{product}}; \mathbf{e}_n) + H(y^{rxn}|\mathbf{e}_n) - H(y^{rxn}|\mathbf{e}_n; \mathbf{z}_{\text{product}}).
\end{aligned} \tag{11}$$

Since $\mathbf{e}_n$ is task-irrelevant noise independent of $y^{rxn}$, we have $H(y^{rxn}|\mathbf{e}_n) = H(y^{rxn})$ and $H(y^{rxn}|\mathbf{e}_n; \mathbf{z}_{\text{product}}) \leq H(y^{rxn}|\mathbf{z}_{\text{product}})$. Then, we have:

$$\begin{aligned}
I(\mathbf{z}_{\text{product}}; \mathbf{e}_{\text{product}}) &\geq I(\mathbf{z}_{\text{product}}; \mathbf{e}_n) + H(y^{rxn}|\mathbf{e}_n) - H(y^{rxn}|\mathbf{e}_n; \mathbf{z}_{\text{product}}) \\
&\geq I(\mathbf{z}_{\text{product}}; \mathbf{e}_n) + H(y^{rxn}) - H(y^{rxn}|\mathbf{z}_{\text{product}}) \\
&= I(\mathbf{z}_{\text{product}}; \mathbf{e}_n) + I(\mathbf{z}_{\text{product}}; y^{rxn}).
\end{aligned} \tag{12}$$

Finally, we obtain $I(\mathbf{z}_{\text{product}}; \mathbf{e}_n) \leq I(\mathbf{z}_{\text{product}}; \mathbf{e}_{\text{product}}) - I(\mathbf{z}_{\text{product}}; y^{rxn})$.

## D   LLM USAGE

Large Language Models (LLMs) were used only for polishing the writing of this paper.

