# OpenReview forum: "Learning Chemical Knowledge from Large-Scale Unlabeled Molecular Data for Retrosynthesis"
_ICLR.cc/2026/Conference — Submitted to ICLR 2026_

### Official Review · Reviewer_2rUs · 2025-10-26

**Soundness:** 2
**Presentation:** 2
**Contribution:** 2
**Rating:** 4
**Confidence:** 5

**Summary:**

This paper leverages large-scale unlabeled molecules to improve the accuracy and diversity of retrosynthesis. It first builds a hierarchical chemical knowledge graph via BRICS decomposition to capture relationships among molecules, substructures, and functional groups. Two layers of R-GCN are then pre-trained on this graph. A task-relevant knowledge adapter extracts effective task-specific KG embeddings, while the retrosynthesis encoder–decoder is pre-trained on a molecule→substructures objective. Finally, the task-relevant KG embeddings are integrated into the pretrained encoder–decoder for end-to-end retrosynthesis training.

**Strengths:**

1. The paper effectively injects chemical knowledge into a hierarchical BRICS-based knowledge graph, explicitly encoding molecule–substructure–functional group relations. Pre-training a two-layer R-GCN on this graph yields stronger, transferable embeddings that provide a better initialization for downstream tasks. A task-relevant adapter further aligns these embeddings with retrosynthesis objectives. In parallel, molecule→substructures pre-training for the encoder–decoder supplies fine-grained supervision, boosting downstream retrosynthesis performance.

2. As shown in Table 1, the approach achieves state-of-the-art performance; Table 3 further demonstrates that every component contributes meaningfully.

**Weaknesses:**

1. Given that the proposed pre-training effectively boosts downstream performance, why restrict the evaluation to single-step retrosynthesis? It would be valuable to report results on multi-step retrosynthesis as well. The approach should also extend naturally to single-step forward reaction prediction and molecular property prediction.

2. The interdependencies between modules are insufficiently articulated, giving the impression of a composite system lacking a unifying design rationale.

**Questions:**

N/A

---

> ### Author Response · Authors · 2025-11-22
>
> Thanks for your detailed review and constructive comments. We address the concerns below.
>
> > **W1:** Given that the proposed pre-training effectively boosts downstream performance, why restrict the evaluation to single-step retrosynthesis? It would be valuable to report results on multi-step retrosynthesis as well. The approach should also extend naturally to single-step forward reaction prediction and molecular property prediction.
>
> **Response:** Thanks for your comments. Although KnowRetro is designed for single-step retrosynthesis, the additional experiments on multi-step retrosynthesis, forward reaction prediction, and molecular property prediction **demonstrate its strong generalizability across broader downstream tasks**.
>
> (1) **Multi-step retrosynthesis**: We incorporated KnowRetro into the search-based planner Retro* and evaluated on the USPTO-190 test set. As shown in Table 1, KnowRetro significantly improves route success rates under various model call settings, demonstrating its utility in Multi-step synthesis planning.
>
> **Table 1.** Multi-step retrosynthesis (route success rate, %) on USPTO-190 with Retro*.
> | Single-Step Model + Planner | 100 Calls | 200 Calls | 500 Calls |
> |----------------|--------------------|--------------------|--------------------|
> | LocalRetro + Retro*         | 58.94              | 64.73              | 73.68              |
> | MEGAN + Retro* | 60.52              | 62.10              | 73.15              |
> | **KnowRetro** + Retro*  | **86.84**          | **94.21**          | **98.95**          |
>
> (2) **Forward reaction prediction**: We tested KnowRetro on the USPTO-MIT-Mixed forward reaction benchmark. Table 2 shows that KnowRetro achieves higher top-k accuracy than the R-SMILES baseline, highlighting the transferability of KnowRetro.
>
> **Table 2.** Forward prediction accuracy (%) on USPTO-MIT-Mixed. “Mixed” indicates that reagents are merged with reactants.
> | Model      | Top-1 | Top-3 | Top-5 | Top-10 |
> |------------|-------|-------|-------|--------|
> | R-SMILES    | 90.0  | 95.6  | 96.4  | 97.2   |
> | **KnowRetro**  | **91.2** | **95.7** | **96.9** | **97.3** |
>
> (3) **Molecular property prediction**: We tested KnowRetro on MoleculeNet tasks following KPGT settings. Table 3 shows that KnowRetro performs competitively on both classification and regression tasks, confirming the general utility of its hierarchical knowledge representations.
>
> **Table 3.** Molecular property prediction on MoleculeNet. Classification: ROC-AUC(higher is better); Regression: RMSE(lower is better).
> | Model          | BBBP | BACE | Tox21 | FreeSolv (RMSE) | ESOL (RMSE) |
> |-------|------|------|-------|-------|-------|
> | GEM [1]     | 0.895 | 0.857 | 0.832 | 2.389 | 0.803 |
> | KPGT [2]  | 0.908 | 0.855 | 0.848 | 2.121 | 0.803 |
> | KnowRetro  | 0.915 | 0.857 | 0.854 | 2.037 | 0.786 |
>
> These results confirm that KnowRetro generalizes effectively across different tasks and settings, highlighting the broader applicability of its knowledge-centric design.
>
> > **W2:** The interdependencies between modules are insufficiently articulated, giving the impression of a composite system lacking a unifying design rationale.
>
> **Response:** KnowRetro consists of three connected stages: \
> (1) **Hierarchical Chemical Knowledge Representation (Fig. 2a):** From large-scale unlabeled molecules, we extract substructures and functional groups to build a multi-level chemical knowledge graph, encoding general transformation-relevant patterns across molecules. \
> (2) **Reaction-Aware Pretraining (Fig. 2b):** The SMILES-to-substructure translation task encourages the encoder to learn fragment-level reactivity patterns. \
> (3) **Knowledge-Guided Retrosynthesis (Fig. 2c):** For a given product, relevant knowledge retrieved from (1) is filtered through a task-relevant adapter and injected into the pretrained encoder from (2). The resulting knowledge-augmented representation conditions the decoder to generate the reactants.
>
> These steps form a clear, end-to-end pipeline in which knowledge flows from construction to pretraining and then to prediction. We have revised Figure 2 and Section 3 to more clearly reflect this unified knowledge-driven framework.
>
> [1] Fang X, Liu L, Lei J, et al. Geometry-enhanced molecular representation learning for property prediction[J]. Nature Machine Intelligence, 2022, 4(2): 127-134.\
> [2] Li H, Zhang R, Min Y, et al. A knowledge-guided pre-training framework for improving molecular representation learning[J]. Nature Communications, 2023, 14(1): 7568.
>
> We sincerely appreciate your thoughtful suggestions, which will help us further strengthen the impact of our work. We hope our responses have effectively addressed your concerns and provided the necessary evidence to support a more favorable assessment.

---

> > ### Comment · Reviewer_2rUs · 2025-11-28
> >
> > Thank you for the response.
> >
> > Some of my concerns are addressed. But in my opinion, this paper is a borderline paper. I leave the decision to AC. It can be accepted or rejected.

---

> > > ### Author Response · Authors · 2025-12-03
> > >
> > > Thanks for the follow-up comment. We sincerely appreciate the time you have invested in reviewing our work.
> > >
> > > In this work, we introduce the **first framework** that learns chemically meaningful hierarchical structure directly from **large-scale unlabeled molecular data**, effectively addressing the scarcity and bias inherent in reaction datasets for single-step retrosynthesis. This design is supported by a comprehensive set of experiments in the paper, all of which consistently validate its effectiveness.
> > >
> > > Moreover, this knowledge-centric design also naturally extends beyond the single-step setting. As suggested, we added experiments on **multi-step planning, forward reaction prediction, and molecular property prediction**, and these additional results demonstrate the **strong generalization ability** of KnowRetro across diverse downstream tasks.
> > >
> > > We believe these contributions, together with the solid empirical evidence, support the significance of KnowRetro.

---

### Official Review · Reviewer_HJdF · 2025-10-30

**Soundness:** 3
**Presentation:** 3
**Contribution:** 3
**Rating:** 6
**Confidence:** 5

**Summary:**

The paper introduces KnowRetro, which constructs a chemical knowledge graph (KG), pretrains using a reaction-aware task (SMILES to BRICS fragments), and injects task-relevant knowledge via a KG adapter for one-step retrosynthesis. It reports SOTA performance on the USPTO-50K dataset.

**Strengths:**

1. This work leverages unlabeled data to construct a hierarchical chemical KG and uses it to guide the generation process.

2. It employs fragment-aware pretraining to enhance performance on the retrosynthesis task.

3. The paper includes detailed ablation experiments.

**Weaknesses:**

1. Negative KG sampling distinguishes between “not in KG” and truly chemically infeasible reactions, which may limit the model’s ability to avoid invalid chemistry.

2. There is a heavy reliance on BRICS/SMARTS rules; the limited coverage and inherent bias of rule-based fragments/functional groups may restrict generalization.

**Questions:**

- Since products are cut into BRICS fragments via predefined rules, could this introduce unintended overlap with ground-truth reactants? Could you quantify any potential leakage by reporting the exact/partial hit rates of main reactants (fragment set vs. ground-truth reactant set) on the test split?

---

> ### Author Response · Authors · 2025-11-22
>
> Thanks for the positive feedback and insightful comments, and suggestions. Our responses are provided below.
>
> > **W1:** Negative KG sampling distinguishes between “not in KG” and truly chemically infeasible reactions, which may limit the model’s ability to avoid invalid chemistry.
>
> **Response:** We construct a stricter negative set by (1) restricting negatives to the same entity type and (2) applying RDKit-based chemical validity checks (e.g., substructure incompatibility) to ensure that sampled negatives reflect truly infeasible. As shown in Table 1, the performance under this stricter setting is very close to that of random negative sampling, suggesting that the distinction between “not in KG” and infeasible reactions has little impact on training in our settings.
>
> **Table 1.** Performance comparison under different negative sampling strategies.
>
> | Negative Sampling Strategy                | Top-1 | Top-3 | Top-5 | Top-10 |
> |-------------------------------------------|-------|-------|-------|--------|
> | Random Negative Sampling         | 62.7  | 82.1  | 88.1  | 92.7   |
> | Strict Negative Sampling   | 63.2  | 82.2  | 89.0  | 91.7   |
>
> > **W2:** There is a heavy reliance on BRICS/SMARTS rules; the limited coverage and inherent bias of rule-based fragments/functional groups may restrict generalization.
>
> **Response:** KnowRetro is not constrained by the potential bias of BRICS/SMARTS rules. We evaluated a variant of KnowRetro using Principal Subgraph Mining (PSM) [1], a data-driven decomposition method that identifies frequent substructures without relying on predefined chemical rules. As shown in Table 2, the BRICS- and PSM-based variants achieve nearly identical performance. This demonstrates that KnowRetro generalizes well across different fragmentation strategies and does not depend on rule-based heuristics, alleviating concerns that BRICS/SMARTS bias limits model generalization.
>
> **Table 2.** Performance under different molecular decomposition strategies.
>
> | Model Variant                     | Top-1 | Top-3 | Top-5 | Top-10 |
> |----------------------------------------------|--------|--------|--------|---------|
> | KnowRetro w/o PT (BRICS)                     | 60.4  | 81.3  | 87.3  | 92.2   |
> | KnowRetro w/o PT (PSM) | 60.1  | 81.5  | 87.4  | 91.9   |
>
> > **Q1:** Since products are cut into BRICS fragments via predefined rules, could this introduce unintended overlap with ground-truth reactants? Could you quantify any potential leakage by reporting the exact/partial hit rates of main reactants (fragment set vs. ground-truth reactant set) on the test split?
>
> **Response:** We believe this overlap does not constitute information leakage. In retrosynthesis, reactants are derived by disconnecting specific bonds in the product and completing the resulting fragments into chemically valid molecules. Thus, some structural similarity between product fragments and reactants is expected and reflects the nature of the task, not unintended leakage.
>
> To assess this empirically, we computed the exact match rate between BRICS-derived product fragments and the main reactants of ground-truth on the USPTO-50K test set (5,007 samples). The match rate is 1.4%, suggesting that such direct overlaps are extremely rare.
>
> To further examine whether these rare overlaps influence model evaluation, we removed all overlapping cases and re-evaluated KnowRetro. As shown in Table 3, the top-k accuracy changes by less than 1%, confirming that these overlaps have little effect on evaluation.
>
> **Table 3.** Accuracy of KnowRetro before and after fragment-reactant overlap filtering.
> | Metric  | Top-1 | Top-3 | Top-5 | Top-10 |
> |---------|-------|-------|-------|--------|
> | KnowRetro | 62.7 | 82.1 | 88.1 | 92.7  |
> | KnowRetro (no overlaps) | 63.2 | 82.8 | 88.8 | 93.3  |
>
> [1] Kong X, Huang W, Tan Z, et al. Molecule generation by principal subgraph mining and assembling[J]. NeurIPS 2022.
>
> We sincerely appreciate your engagement and hope our response has addressed the concerns to support a more favorable evaluation.

---

### Official Review · Reviewer_8eRW · 2025-10-31

**Soundness:** 3
**Presentation:** 2
**Contribution:** 3
**Rating:** 4
**Confidence:** 4

**Summary:**

This work investigates inorganic retrosynthesis by constructing a hierarchical knowledge graph (KG) that incorporates fine-grained substructures and functional groups. The authors further apply the Information Bottleneck (IB) principle to capture task-relevant knowledge. Various experiments are conducted to demonstrate the superiority of the proposed method, KnowRetro.

**Strengths:**

- Compared to previous methods that only model KG relationships between reactants and products, this work builds a hierarchical KG leveraging more fine-grained information at the substructure and functional-group levels, and demonstrates its effectiveness through diverse experiments including ablation studies.
- Extensive experiments, including a theoretical analysis, are provided.

**Weaknesses:**

- The overall structure of the paper does not clearly and systemically explain how all components of the method connect to each other.
Beyond the theoretical analysis, further explanation and experimental validation are needed to justify the necessity and effectiveness of the Task-Relevant Knowledge Adapter.
It remains unclear whether the model actually achieves significant information gain for retrosynthesis compared to directly using the raw KG.
- In Equation (9), the process of penalizing “excessive dependence” on e_product needs more clarification — what exactly constitutes “excessive dependence”?
In retrosynthesis, the input embeddings arguably contain the most critical information, so it is unclear why penalizing their dependence would be beneficial.
- It would also strengthen the paper to include a specific example showing how the Information Bottleneck filtering removes noisy or irrelevant information from the original KG.
- While the Reliability of KnowRetro on Noisy KGs experiment is valuable, it only covers artificially induced noise; showing examples of real-world KG noise and how the filtering mitigates it would be more convincing.
Additionally, comparisons against baselines on other datasets such as Pistachio or ORD would improve the completeness of the evaluation.
- Although an anonymized GitHub link is provided, the code could not be accessed.

**Questions:**

See Weakness section above.

---

> ### Author Response · Authors · 2025-11-22
> **Response to Reviewer 8eRW (Part 1)**
>
> Thanks for taking the time to carefully review our work and for the constructive suggestions. We address each comment below.
>
> > **W1.1:** Explain how all components of the method connect to each other.
>
> **Response:** KnowRetro contains three modules that interact in an end-to-end manner: \
> (1) **Hierarchical Chemical Knowledge Representation (Fig. 2a; Sec. 3.1):** Large-scale unlabeled molecules are decomposed into substructures and functional groups to build a multi-level knowledge graph, which is then encoded to enable retrieval of product-specific chemical knowledge; \
> (2) **Reaction-Aware Pretraining (Fig. 2b; Sec. 3.2)**: An encoder–decoder is pretrained with a SMILES-to-substructure objective, allowing the encoder to learn fragment-level reactivity patterns; \
> (3) **Knowledge-guided Retrosynthesis Prediction (Fig. 2c; Sec. 3.3)**: Retrieved product knowledge from (1) is distilled by a task-relevant adapter and then used to condition the reaction-aware encoder from (2). The resulting knowledge-augmented representation then guides the decoder to generate the reactants.
>
> We have revised Figure 2 and the Overview section to more explicitly show how these components connect across the pipeline.
>
> > **W1.2:** Necessity and Effectiveness of the Task-Relevant Knowledge Adapter.
>
> **Response:** We demonstrate the necessity and effectiveness of the adapter through both performance and representation analyses (Appendix B.7). The adapter **consistently improves** prediction accuracy and **produces clearer** reaction-level representations, indicating that it plays a critical role in our framework. Specifically:
> - **Performance improvements.** As shown in Table 12, adding the adapter improves all Top-k metrics compared with directly injecting raw KG embeddings (2% relative gain in Top-1).
> - **More discriminative representations.** Figure 7 shows that reaction embeddings become more compact and better separated across major reaction classes, even **without reaction-type supervision**, indicating that the adapter makes reaction-relevant knowledge more discriminative and reduces the influence of KG components that are less helpful for identifying the transformation.
>
> These results show that the adapter provides stable improvements in both predictive accuracy and representation quality, confirming that the task-relevant adapter is both necessary and effective.
>
> > **W2:** In Eq. (9), the idea of penalizing “excessive dependence” on $e _ {\text{product}}$ is unclear. Since the product embedding carries essential information for retrosynthesis, why would reducing dependence on it be beneficial?
>
> **Response:** In Eq. (9), penalizing “excessive dependence” does not mean reducing essential information in the product embedding. Instead, the product embedding contains many frequent but low-informative patterns, for example aromatic rings or distal substituents, that do not guide the model toward the correct disconnection site. If the model relies too heavily on these signals, it becomes less sensitive to the transformation-relevant parts of the product. The IB-based adapter, through the KL term in Eq. (9), encourages $z _ {\text{product}}$ to keep only the information from $e _ {\text{product}}$ that is useful for predicting the retrosynthetic target, while compressing away redundant dependence on such common patterns. The coefficient $\beta$ controls the strength of this bottleneck (larger $\beta$ = stronger compression), and Table 13 (Appendix B.8) shows that a small but non-zero $\beta$ gives the best accuracy, indicating that selectively filtering out these irrelevant signals indeed improves retrosynthesis predictions.

---

> > ### Author Response · Authors · 2025-11-22
> > **Response to Reviewer 8eRW (Part 2)**
> >
> > > **W3:** It would also strengthen the paper to include a specific example showing how the Information Bottleneck filtering removes noisy or irrelevant information from the original KG.
> >
> > **Response:** To provide a concrete example of how the Information Bottleneck (IB) adapter filters noisy or irrelevant KG information, we consider the following thiourea-forming reaction (US08748426B2):
> > - **Reaction:** `FC(F)(F)c1ccccc1N=C=S.COC(=O)[C@H](C(C)C)N1Cc2cc(-c3ccc(N)cc3)ccc2C1=O`>>`COC(=O)[C@H](C(C)C)N1Cc2cc(-c3ccc(NC(=S)Nc4ccccc4C(F)(F)F)cc3)ccc2C1=O`
> > - **Product Fragments**: phenyl (`c1ccccc1`), methoxy (`CO`), isoindolinone (`O=C1NCc2ccccc21`), trifluoromethyl (`FC(F)F`), aliphatic acyl chain (`CC(C)CC=O`), and thiourea (`NC(N)=S`).
> >
> > Among these fragments, `NC(N)=S` is the only transformation-relevant unit. The others do not participate in the transformation but appear far more frequently in the KG (e.g., `FC(F)F` occurs 56,182 times vs. 2,235 for `NC(N)=S`), which biases the raw product embedding toward these common patterns and gives them higher similarity (i.e., cosine similarity). After applying the IB adapter, similarity shifts away from these high-frequency irrelevant fragments and toward the true thiourea unit, as shown in Table 1.
> >
> > **Table 1.** Nearest KG Fragments before and after the IB-based adapter.
> >
> > | Fragments | Description            | Adapter-Before | Adapter-After |
> > |---------------------|------------------------|---------------------|---------------------|
> > | `O=C1NCc2ccccc21`     | isoindolinone scaffold | 0.64 (Rank 1) | 0.31           |
> > | `FC(F)F`                          | trifluoromethyl group  | 0.39 (Rank 2)   | 0.19           |
> > | `CC(C)CC=O`                       | aliphatic acyl chain   | 0.26(Rank 3)            | 0.09           |
> > | `CO`                              | methoxy group          | 0.18(Rank 4)              | 0.04           |
> > | **`NC(N)=S`**                     | **thiourea**           | **0.06** (Rank 5)          | **0.59** (Rank 1)    |
> > | `c1ccccc1`                       | phenyl            | 0.01 (Rank 6)            | 0.06           |
> >
> > This example shows that the IB adapter reduces the influence of frequent but irrelevant KG fragments and strengthens the signal of the true reaction-forming unit, so that the product representation better reflects the actual transformation.
> >
> > > **W4.1:** While the Reliability of KnowRetro on Noisy KGs experiment is valuable, it only covers artificially induced noise; showing examples of real-world KG noise and how the filtering mitigates it would be more convincing.
> >
> > **Response:** A common source of real-world KG noise comes from functional-group misannotation, where different groups share similar local substructures and are therefore assigned closely related but incorrect functional-group labels. For example, in the following carbamate-forming reaction (US20060160832A1):
> > - Reaction: `O=C(Cl)Oc1ccccc1.COc1cc2nccc(Oc3ccc(N)cc3)c2cc1OC` >> `COc1cc2nccc(Oc3ccc(NC(=O)Oc4ccccc4)cc3)c2cc1OC`
> >
> > the newly formed reaction center is a carbamate group `OC(=O)NH`, but it may be incorrectly labeled as a generic amide `C(=O)N`, which is a typical real-world misannotation in the KG.
> >
> > To examine how the adapter handles this type of noise, we compare the cosine similarity between the product embedding and functional-group embeddings before and after the adapter. As shown in Table 2 below, before the adapter, generic scaffold and amide features receive higher similarity scores than the true carbamate node. After the adapter, the carbamate `OC(=O)NH` becomes the top-scoring functional group, while the incorrect amide `C(=O)N` is clearly down-weighted.
> >
> > **Table 2.** Example of carbamate–amide misannotation and the effect of the adapter.
> > | Functional group | Description                | Adapter-Before        | Adapter-After         |
> > |------------------|---------------------------|------------------------|-----------------------|
> > | `c1ccc2ncccc2c1`   | quinoline scaffold        | 0.43 (Rank 1)          | 0.15          |
> > | `C(=O)N`           | amide                     | 0.24 (Rank 2)          | 0.22          |
> > | `c1ccccc1`         | phenyl ring               | 0.21 (Rank 3)          | 0.12          |
> > | `CO`                | methoxy group             | 0.11 (Rank 4)          | 0.05          |
> > | **`OC(=O)NH`**     | **carbamate (reaction center)** | **0.08 (Rank 5)** | **0.34 (Rank 1)**     |
> >
> > This example illustrates that the adapter reduces the impact of misannotated labels and reinforces the true reaction center, effectively mitigating real-world KG noise. These observations are consistent with our reliability experiments in Section 4.4 (Fig. 4).

---

> > > ### Author Response · Authors · 2025-11-22
> > > **Response to Reviewer 8eRW (Part 3)**
> > >
> > > > **W4.2:** Additionally, comparisons against baselines on other datasets such as Pistachio or ORD would improve the completeness of the evaluation.
> > >
> > > **Response:** USPTO-50K (50k+) and USPTO-FULL (950k+) are widely adopted benchmarks in retrosynthesis, and our main experiments follow this standard to ensure fair and consistent comparison with previous work.
> > >
> > > As recommended, we further evaluate KnowRetro on the ORD dataset (Train/Val/Test: 1,819,522 / 227,440 / 227,440), since Pistachio is not publicly accessible. This allows for a reproducible cross-dataset evaluation. As shown in Table 3, KnowRetro consistently outperforms R‑SMILES, a sequence-based baseline that does not incorporate external chemical knowledge. Notably, both models were trained under the same data setting, without any data augmentation. This further confirms the robustness and generalizability of our knowledge-guided framework.
> > >
> > > **Table 3.** Evaluation on the ORD dataset (no data augmentation).
> > > | Model      | Top-1 | Top-3 | Top-5 | Top-10 |
> > > |------------|-------|-------|-------|--------|
> > > | R-SMILES   | 33.28 | 44.45 | 48.12 | 51.43  |
> > > | **KnowRetro** | **37.87** | **46.73** | **51.97** | **58.11** |
> > >
> > >
> > > > **W5:** Access issue with the anonymized link.
> > >
> > > **Response:** We have verified that the anonymized GitHub link is accessible. Access issues may occur if the link is copied together with punctuation at the end. If convenient, could you please try again with the URL as provided? If the problem persists, please let us know, and we would be happy to provide additional details.
> > >
> > > We sincerely appreciate your constructive feedback and hope our responses address your concerns and contribute to a more favorable evaluation.

---

> ### Comment · Reviewer_8eRW · 2025-11-28
>
> Thank you for the rebuttal and for providing additional examples. However, since the main novelty lies in leveraging the hierarchical knowledge graph, I find the performance gains relatively limited, and the paper lacks sufficient analysis demonstrating how the hierarchical structure—beyond a standard knowledge graph—contributes to the improvements.
>
> I will keep my current score, and I remain neutral regarding acceptance.

---

> > ### Author Response · Authors · 2025-12-03
> >
> > Thanks for the comment. We would like to clarify that our work does not simply `leverage a hierarchical knowledge graph`. Instead, we propose an innovative retrosynthesis framework that learns chemically meaningful hierarchical structure directly from **large-scale unlabeled molecular data**, addressing the scarcity and bias of reaction datasets. We also clarify the additional information raised by the reviewer:
> > - **Performance gains are consistent across datasets and metrics.** KnowRetro improves Top-10 performance on USPTO-50K (Unknown: **92.7% vs. 90.3%** for EditRetro (data-driven method) and **86.8%** for PMSR (knowledge enhanced method); Known: **96.8% vs. 92.7%** for PMSR). The improvement also holds on the more diverse USPTO-FULL (**80.1% vs. 70.1%** for PMSR and 74.2% for EditRetro). Beyond accuracy, KnowRetro achieves the best MaxFrag (Top-10: **94.1% vs. 92.8%**) and stronger Round-Trip validity (Top-10: **62.0% vs. 50.8%**), showing consistent gains not only in correctness but also chemical validity and diversity.
> > - **Ablation studies directly demonstrate that the hierarchical structure provides benefits beyond a standard (molecule-level) KG.** As shown in Table 3 (Sec. 4.3, revised manuscript), the standard KG performs worst (**56.5%**), providing little guidance for retrosynthesis. Adding substructures (60.6%) or functional groups (61.9%) brings clear improvements, and combining all levels achieves the best performance (62.7%). This consistent progression demonstrates that each level contributes complementary chemical information that single-level, standard KGs cannot provide.
> > - **Attention analysis reveals how hierarchical knowledge changes model behavior.** As shown in Sec. 4.5 and Appendix B.11, the variant without KG attends mainly to SMILES syntax rather than chemically relevant regions. In contrast, KnowRetro consistently focuses on reaction centers (such as O–C or N–C cleavages) and the structural fragments directly attached to them. This indicates that hierarchical knowledge enables transfer of local reactivity patterns learned from molecular data, leading to chemically plausible disconnections.
> >
> > We hope these analyses clarify how the hierarchical structure contributes to improvements beyond a standard KG and highlight our contribution of enabling retrosynthesis models to effectively leverage large-scale unlabeled molecular data, which is especially valuable given the scarcity of publicly available high-quality reaction datasets.

---

### Author Response · Authors · 2025-11-27
**A Follow-up Before the Discussion Ends**

Dear Reviewers,

Thank you again for your thoughtful and constructive feedback. As the discussion phase comes to an end, we would greatly appreciate it if you could let us know whether our responses and the additional experiments have fully addressed your comments. We are happy to clarify any remaining points during the rest of the discussion period.

We truly appreciate your time and consideration.

Best regards,
Authors

---

### Author Response · Authors · 2025-12-03
**Final Remarks by Authors**

Dear (Senior) AC and Reviewers,

**We sincerely thank you for the time and effort dedicated to evaluating our manuscript.** To facilitate your efficient assessment, we concisely summarize the key strengths, concerns, and our corresponding revisions across all reviews below.

| Reviewer | Score | Strengths / Acknowledgment | Weaknesses / Concerns | Our Response / Revision |
| -------- | ----- | ---------------------- | ---------------------- | ---------------------- |
| 8eRW     | 4     | Sound method with theoretical analysis; strong empirical results |  Figure unclear;  lack of additional examples | Revised the model figure; added intuitive examples clarifying KG denoising  |
| HJdF     | 6     | Strong motivation for learning from unlabeled data; solid experimental results  | Need more discussion on negative KG sampling and BRICS rules | Added stricter negative-sampling experiments and data-driven PSM decomposition, further validating KnowRetro's robustness |
| 2rUs     | 4     | Method effectiveness; strong results              | Need broader task evaluation beyond single-step retrosynthesis                              | Added multi-step, forward reaction, and molecular property prediction experiments further demonstrating KnowRetro's generalization |

**Overall**, **all reviewers acknowledge that the method is effective, technically sound, and well-supported by extensive experiments**, including ablations, theoretical analysis, and additional studies provided during rebuttal. The remaining concerns focused primarily on **presentation clarity** and the need for **additional illustrative examples** (Reviewer `8eRW`), and **broader task evaluation** (Reviewer `2rUs`).

We have fully addressed these points by incorporating:
- Improved presentation & clarity (Sec. 3 Overview, Fig. 2)
- Clearer explanations with real-world examples
- Additional experiments and broader downstream evaluations (Fig. 5, Fig. 6, and Table 3)

**All corresponding revisions have been integrated into the revised manuscript.**

Due to rebuttal policy constraints, we were unable to receive further feedback from the reviewers after submitting these detailed clarifications. We respectfully ask the AC to consider the substantial revisions and additions we made directly in response to the reviewers’ suggestions.

Thank you again for your thoughtful evaluation.

Best regards,

Authors

---

### Meta-Review · Area_Chair_9uvn · 2026-01-07

**Summary:**

KnowRetro learns a hierarchical chemical KG from unlabeled molecules, pre-trains encoder-decoder with fragment-level tasks, and injects task-relevant knowledge via an IB adapter for single-step retrosynthesis. Reviewers praised the idea and strong USPTO results, but worried about (i) limited gain vs. standard KG, (ii) rule-based BRICS bias, (iii) narrow single-task evaluation, (iv) unclear module interaction and (v) noisy-KG robustness evidence. Authors supplied new multi-step, forward-reaction and property-prediction experiments, ablations showing each hierarchy level helps, PSM decomposition test, stricter negative-sampling, real-world mis-annotation examples, and revised figures.

**Reviewer Concerns:**

Addressed: broader tasks, ablation on hierarchy, PSM vs BRICS, stricter negatives, real noise example, module connectivity, code link checked.
Still outstanding: only marginal absolute Top-1 gain over strong baselines; no public Pistachio test; oral-level novelty not fully convinced.

**Reviewer Scores:**

R1  4
R2  6
R3  4

---

### Decision · Program_Chairs · 2026-01-26

Reject